# Competitively Consistent Clustering

**Niv Buchbinder**[* 1]   **Roie Levin**[* 2]   **Yue Yang**[* 2]

## Abstract

In *fully-dynamic consistent clustering*, we are given a finite metric space $(M, d)$, and a set $F \subseteq M$ of possible locations for opening centers. Data points arrive and depart, and the goal is to maintain an approximately optimal clustering solution at all times while minimizing the *recourse*, the total number of additions/deletions of centers over time. Specifically, we study fully dynamic versions of the classical $k$-center, facility location, and $k$-median problems. We design algorithms that, given a parameter $\beta \geq 1$, maintain an $O(\beta)$-approximate solution at all times, and whose total recourse is bounded by $O(\log |F| \log \Delta) \cdot \text{OPT}_{\text{REC}}^{\beta}$. Here $\text{OPT}_{\text{REC}}^{\beta}$ is the minimal recourse of an offline algorithm that maintains a $\beta$-approximate solution at all times, and $\Delta$ is the metric aspect ratio. Finally, while we compare the performance of our algorithms to an optimal solution that maintains $k$ centers, our algorithms are allowed to use slightly more than $k$ centers. We obtain our results via a reduction to the recently proposed *Positive Body Chasing* framework of [Bhattacharya, Buchbinder, Levin, Saranurak, FOCS 2023], which we show gives fractional solutions to our clustering problems online. Our contribution is to round these fractional solutions while preserving the approximation and recourse guarantees. We complement our positive results with logarithmic lower bounds which show that our bounds are nearly tight.

## 1. Introduction

Clustering is a fundamental optimization primitive and one of the most widely used tools in data analysis. Given a dataset in a metric space, the task is to output a set of cluster *centers* that minimize an objective which is a function of the distances between data points and their nearest center. In this work we study three such clustering formulations: (i) *$k$-center*, in which we can open at most $k$ centers and we seek to minimize the maximum distance of any point to its nearest open center, (ii) *facility location*, in which there is cost to open each center, and we wish to minimize the sum of opening costs plus the sum of distances between each data points and its nearest open center, and (iii) *$k$-median*, in which we can open at most $k$ centers, and we wish to minimize the sum of distances between each data point and its nearest open center. These classical objectives have been studied extensively for the last few decades, and though they are NP-hard, their approximability is almost completely understood (Byrka et al., 2017; Byrka & Aardal, 2010; Jain et al., 2002; Hochbaum & Shmoys, 1986; Guha & Khuller, 1999).

In practice however, we are often faced with situations where the data is not static but evolving over time. As the data change, it may be important to maintain a set of cluster centers that not only minimizes the objective at hand, but also does not change drastically between time steps. For example, suppose we are clustering as a preprocessing step to split data among servers; shuffling data between servers with every update could be prohibitively costly. This harder task, known as *consistent clustering*, is less well understood than its offline counterpart and has drawn significant attention in recent years from both academic and industry researchers (Lattanzi & Vassilvitskii, 2017; Cohen-Addad et al., 2022; 2019; Chan et al., 2024; Bhattacharya et al., 2022a; Fichtenberger et al., 2021; Łącki et al., 2024; Guo et al., 2021; 2020; Moseley et al., 2023; Bhattacharya et al., 2024b; Forster & Skarlatos, 2025; Bhattacharya et al., 2024a).

To the best of our knowledge, all state-of-the-art fully-dynamic consistent clustering results aim for *absolute recourse* bounds. These are guarantees of the form "After $T$ data point insertions or deletions, the algorithm incurs at most $c \cdot T$ recourse." Consider, for example, the recent work of Łącki at el. (Łącki et al., 2024) which gives an algorithm for fully-dynamic $k$-center that maintains a constant approximation with $O(T)$ recourse. This highly non-trivial bound is a significant improvement over previous results (Lattanzi

---

[*]Authors ordered alphabetically. [1]Department of Statistics and Operations Research, School of Mathematical Sciences, Tel Aviv University, Israel. [2]Department of Computer Science, Rutgers University, Piscataway, NJ 08854. Correspondence to: Niv Buchbinder <niv.buchbinder@gmail.com>, Roie Levin <roie.levin@rutgers.edu>.

*Proceedings of the 42ⁿᵈ International Conference on Machine Learning*, Vancouver, Canada. PMLR 267, 2025. Copyright 2025 by the author(s).

& Vassilvitskii, 2017; Fichtenberger et al., 2021); it is also best possible in the sense that there exist update sequences for which $o(T)$ recourse does not suffice to maintain a constant approximation.

However, these sequences in which every algorithm incurs $\Omega(T)$ recourse seem pathological and unrealistic: for example, if a set of $k+1$ points that are far from each other depart then return repeatedly in round-robin fashion, any good $k$-clustering approximation algorithm must open and close a center at every time step. In fact, low-recourse clustering makes the most sense as a goal when the data has consistent structure over time, i.e. precisely when there exists an offline solution that rarely change its centers over time.[1] In recent work, (Bhattacharya et al., 2023) initiated the study of algorithms with *competitive recourse*, which have refined guarantees of the form "Over the course of the input sequence, the algorithm incurs recourse that is at most $c \cdot \text{OPT}_{\text{REC}}$." Here $\text{OPT}_{\text{REC}}$ which is the minimal recourse of *any* offline algorithm with full foreknowledge of the input that maintains an optimal solution at all times. More generally, for any $\beta \geq 1$, they compare the online algorithm's recourse to $\text{OPT}_{\text{REC}}^{\beta}$, which is the optimal recourse of an offline algorithm that is allowed to maintain a $\beta$-approximate solution at all times.[2] To rephrase in this language, we claim that for realistic instances of consistent clustering and reasonable constant values of $\beta$, we have $\text{OPT}_{\text{REC}}^{\beta} \ll T$.[3]

### 1.1. Results

In this work we initiate the study of consistent clustering algorithms in the *fully dynamic* model with competitive recourse. We study fully dynamic versions of three classical clustering problems, the $k$-center problem, the facility location, and the $k$-median problem. Our main result is the following.

**Theorem 1.1** (Upper Bound). *For every $\beta \geq 1$, there exists a dynamic clustering algorithm*

- *for $k$-**center** that given an $\epsilon \in (0, 1/2)$ maintains an $O(\beta)$-approximate solution, uses $(1+\epsilon) \cdot k$ centers and*

*incurs recourse at most $O(\frac{1}{\epsilon^2} \log \frac{|F|}{\epsilon} \log \Delta) \cdot \text{OPT}_{\text{REC}}^{\beta}$.*

- *for **facility location** that maintains an $O(\beta)$-approximate solution and incurs recourse at most $O(\log |F| \log \Delta) \cdot \text{OPT}_{\text{REC}}^{\beta}$.*

- *for $k$-**median** that maintains an $O(\beta)$-approximate solution, uses $O(k)$ centers, and incurs recourse at most $O(\log |F| \log \Delta) \cdot \text{OPT}_{\text{REC}}^{\beta}$.*

*Here $\Delta$ is the aspect ratio of the metric space, and $F$ is the set of locations where the algorithm can open a center.*

We complement our upper bounds by showing that at least one logarithmic factor is unavoidable.

**Theorem 1.2** (Lower Bound). *Suppose there is a randomized algorithm that given a parameter $\beta \geq 1$ as an input maintains an $(\alpha \cdot \beta)$-approximation for one of fully-dynamic facility location, $k$-center, or $k$-median. Then this algorithm has recourse at least $\Omega(\min\{\log |F|, \log_\alpha \Delta\} \cdot \text{OPT}_{\text{REC}}^{\beta}$. For $k$-center and $k$-median, the lower bound holds even if the algorithm is allowed to open $O(k)$ facilities.*

Finally, we evaluate our three clustering algorithms experimentally on UCI Machine Learning repository datasets. Our experiments demonstrate not only that our algorithms are simple to implement in practice, but that they also significantly outperforms the worst-case bound predicted by the theorem. They also seem to show that $\text{OPT}_{\text{REC}}^{\beta} \ll T$ as we predict.

### 1.2. Techniques and overview

The starting point of our work is the positive body chasing framework of (Bhattacharya et al., 2023). Loosely speaking, they show that given a sequence of convex bodies $K_1, K_2, \ldots, K_T$ revealed online, when these bodies are defined by packing and covering constraints, there is an algorithm that maintains a point $x^t \in K_t$ online such that the total $\ell_1$ movement $\sum_t \|x^t - x^{t-1}\|_1$ (or "fractional recourse"), is within a logarithmic factor of $\text{OPT}_{\text{REC}}$, the minimum fractional recourse of an offline algorithm maintaining a point in the same bodies. In Appendix A we show how to cast a suitable fractional version of our three dynamic clustering problems as positive body chasing.

The main contribution of our work is to show how to *round* the fractional solution output by (Bhattacharya et al., 2023) while preserving both recourse and approximation guarantees. Offline, rounding is already a non-trivial task tailored for each problem separately. Online, rounding imposes additional challenges: not only should the integral solution preserve feasibility and the objective function value, but it should also maintain a stable solution over time. The latter property does not hold for most textbook rounding techniques, for which a small change in the fractional solution

---

[1]This is reminiscent of (Awasthi et al., 2012) which gives polynomial time algorithms for clustering instances whose solutions are stable under perturbations. They posit that these are inputs one would realistically want to cluster in the first place.

[2]Note that the value $\text{OPT}_{\text{REC}}^{\beta}$, by definition, decreases as $\beta$ increases. For the $k$-center objective, the proof of (Łącki et al., 2024) shows that there is a setting of $\beta = O(1)$ such that $\text{OPT}_{\text{REC}}^{\beta} = O(T)$.

[3]We note that (Fichtenberger et al., 2021) can be interpreted as a competitive recourse guarantee, but this result is only for the incremental setting. They show an $\tilde{O}(k)$ absolute recourse algorithm, which is also poly log competitive because $k$ is a lower bound on the recourse of any algorithm opening $k$ centers.

may result in large changes to the rounded integral solution. To this end, various online rounding ideas were proposed in the context of other problems (Alon et al., 2009; Bansal et al., 2012; Adamaszek et al., 2012; Bansal et al., 2011; Krishnaswamy et al., 2023).

In Section 3.1 we design a rounding algorithm for the $k$-center problem. It is well known that the greedy algorithm that repeatedly adds to the solution any point farther than $\alpha \cdot$ OPT (for $\alpha \geq 2$) to an open center opens at most $k$ centers, and hence is a $\alpha$-approximation (Gonzalez, 1985). Can we emulate this strategy in the dynamic setting using the fractional solution as our guide? When the maximum intracluster distance OPT does not change, the LP guarantees that every new center OPT far from an open center has an LP mass of 1 lying in the ball of radius OPT around it. If we run the greedy algorithm and, as the fractional solution moves, we also *drop* centers whose ball's mass dips significantly below 1, then we can charge the recourse of the center leaving the solution to the $\ell_1$ movement of the fractional solution that caused the dip.

The story is more involved in the general case when the value of OPT changes over time. When the value of OPT changes, the strategy above may drop balls despite the fractional solution not moving at all, and hence a more careful argument is required. In particular, we would like our algorithm to be "lazy" and only move centers when it has no other choice. Our algorithm actively maintains a set of "well separated" balls of different radii around its chosen cluster centers, and uses a potential function argument (and a set of invariants) to bound the number of balls we drop before the solution reaches feasibility. Intuitively, whenever we drop a ball and the fractional solution does not change enough, we show that a new ball that is added has a much smaller radii compared with the ball we dropped, and this can only happen $O(\log \Delta)$ many times.

Our rounding scheme for facility location in Section 3.2 can be seen as a dynamic analog of the classic result of (Shmoys et al., 1997). Roughly, let $R_j$ be the fractional service cost of a point/client $j$ (we will define this carefully later). By Markov's inequality, for a constant $\gamma \geq 1$, every client $j$ must have a fractional mass of $1 - 1/\gamma$ within the ball of distance $\gamma \cdot R_j$ from $j$. In (Shmoys et al., 1997) they show that greedily picking disjoint balls of this kind in order from smallest to largest radius and opening the cheapest facility within each is a constant approximate solution.

Once again, as the underlying fractional solution changes over time, these service costs change, and we need to be careful with our online choices. As in the $k$-center case, our algorithm maintains a set of "well separated" balls of different radii around certain points/clients and it also chooses a cheap facility inside each such ball. If the total fraction inside a ball drops enough, we can charge our integral re-

course to this change. Otherwise, we show via a potential function argument (and a set of invariants) how to bound the number of balls we drop. Our $k$-median rounding algorithm, which we discuss in Section 3.3, turns out to be a direct consequence of the result in Section 3.2. In Section 4, we evaluate implementations of all our algorithms experimentally.

In Appendix C we show our logarithmic lower bounds that draws ideas from the lower bound in (Fotakis, 2008). The lower bound holds even against fractional algorithms (and hence also randomized algorithms). For all three problems, we use variants of a uniform binary HST metric, and the request sequence inserts clients along a root to leaf path. The adversary adaptively picks this path such that the next request falls in a subtree with very few open centers, fractionally. The nature of the metric ensures that the algorithm must move fractional mass into this subtree if it wants to maintain a good approximation, thus incurring high recourse.

### 1.3. Additional related work

Clustering under various objectives is by now a classic problem with many known approximation algorithms. We mentioned a few besides those included in the introduction (Arya et al., 2004; Kanungo et al., 2004; Jain et al., 2003; Charikar & Guha, 2005), but for a more complete treatment, see the relevant sections of (Williamson & Shmoys, 2011).

Besides the stable clustering algorithms mentioned above, there is a well-established line of work on low-recourse dynamic algorithms for a host of combinatorial problems, e.g. Steiner tree (Imase & Waxman, 1991; Gu et al., 2016; Gupta & Kumar, 2014; Łącki et al., 2015; Gupta & Levin, 2020), load balancing (Awerbuch et al., 2001; Gupta et al., 2014; Krishnaswamy et al., 2023), set cover (Gupta et al., 2017; Abboud et al., 2019; Bhattacharya et al., 2019; Gupta & Levin, 2020; Bhattacharya et al., 2021; Assadi & Solomon, 2021), edge orientation (Brodal & Fagerberg, 1999; Sawlani & Wang, 2020; Bera et al., 2022), graph coloring (Solomon & Wein, 2020), maximal independent sets (Assadi et al., 2018), and spanners (Baswana et al., 2012; Bhattacharya et al., 2022b).

## 2. Preliminaries

In the classical clustering problems we study we are given a metric space $(M, d)$ on $n$ points. We assume without loss of generality that $\min_{i,j \in M} d_{ij} = 1$ such that $\Delta = \max_{i,j \in M} d_{ij}$ is the aspect ratio of the metric space. In addition, there is a set of points $C \subseteq M$ we refer to as clients, and a set of candidate center locations $F \subseteq M$.[4] The algorithm outputs a solution which is a subset of the

---

[4] In $k$-center and $k$-median, we assume $F = M$.

candidate center locations, $S \subseteq F$. The constraints and objectives are different for each of the problems we study:

- In $k$-center, the algorithm is allowed to open at most $k$ centers and the goal is to minimize the maximal distance of a client in $C$ to its closest open center.

- In facility location, the algorithm is allowed to open any number of centers (which we alternatively refer to as a facilities), but opening a facility in position $i \in F$ incurs an opening cost of $f_i$. The goal is to minimize the total opening cost plus the total service cost, defined as the sum of distances from the clients in $C$ to their nearest open facility.

- In $k$-median, the algorithm is allowed to open at most $k$ centers and the goal is to minimize the sum of distances between clients and their closest center.

In the fully dynamic version of each of these problems, we are given at every time $t = 1, \ldots, T$ a different set of clients $C^t \subseteq M$. The algorithm must maintain a solution $S^t$ at each time step that is approximately optimal with respect to the clustering objective at that time step, while also minimizing the total recourse, defined as $\sum_{t=1}^{T} |S^t \oplus S^{t-1}|$. More precisely, our goal in this work is to maintain an $(\alpha \cdot \beta)$-approximate solution to the clustering objective while paying recourse at most $c \cdot \text{OPT}_{\text{REC}}^{\beta}$, where $\text{OPT}_{\text{REC}}^{\beta}$ denotes the optimal recourse of an offline solution that maintains a $\beta$-approximate solution at all times. For each problem, we use $\text{OPT}^t$ to denote the cost of the optimal *fractional* solution at time $t$.

## 3. Algorithms

### 3.1. A rounding algorithm for fully-dynamic $k$-center

In this section we design a rounding algorithm for the fully dynamic $k$-center problem. Following Appendix A.1, for every $\epsilon \in (0, 1]$ there is an algorithm that maintains a fractional solution $x \geq 0$ to the fully dynamic $k$-center problem satisfying the guarantees:

$$\sum_{t=1}^{T} \|x^t - x^{t-1}\|_1 \leq O\left(\frac{\log(\frac{n}{\epsilon})}{\epsilon}\right) \cdot \text{OPT}_{\text{REC}}^{\beta},$$

$$\sum_{i \in B_j^t} x_i^t \geq 1 \qquad \forall j \in C^t, t \in [T],$$

$$\sum_{i=1}^{n} x_i^t \leq (1+\epsilon)k \qquad \forall t \in [T].$$

For an active client $j \in C^t$ and time $t$, $B_j^t = \{i \in M \mid d_{ij} \leq \min\{\beta \cdot OPT^t, \Delta\}\}$ is the set of centers at distance at most $\beta \cdot OPT^t$, and therefore feasibly serve client $j$ at time $t$, and let $D^t = \min\{\beta \cdot OPT^t, \Delta\}$. The bound in Theorem

1.1 for the fully dynamic $k$-center problem is obtained by combining the guarantee on the fractional solution along with the following theorem that we prove.

**Theorem 3.1** (Dynamic $k$-center rounding). *Let $x^t$ be a fractional solution to the dynamic $k$-center problem that maintains a solution that is a $\beta$-approximation at any time, uses at most $k'$ centers, and its total fractional recourse is $R$. Then, for any $\epsilon \leq 1/2$, there is an algorithm that maintains an* **integral** *solution with at most $(1 + 2\epsilon)k'$ centers that is $(\alpha \cdot \beta)$-approximate at any time, and its total recourse is $O\left(\log(\Delta)/\epsilon\right) \cdot R$, where $\alpha = 3 + 2\sqrt{2} \approx 5.28$.*

**The Algorithm:** Our algorithm maintains at time $t$ a set of open centers $S^t$. For each $i \in S^t$, let $t_i \leq t$ be the time in which the center is added to the solution and let $r_i = D^{t_i} = \min\{\beta \cdot OPT^{t_i}, \Delta\}$ be the radius of the balls that can serve the clients when $i$ was added to $S^t$ (note that $r_i \leq \Delta$). With foresight, we set parameters $\alpha = 3 + 2\sqrt{2}$ and $\delta = \sqrt{2}$ and define $B_i := \{j \mid d_{ij} \leq r_i\}$ and $\widehat{B_i^t} := \{j \mid d_{ij} \leq \alpha \cdot D^t\}$. We identify each facility with the ball around it, and count in our analysis the number of balls that we add or drop. Our algorithm is simple to describe and works as follows. At time $t$ we update the fractional solution and the value of $OPT^t$ (the radius of the $k$-center optimal fractional solution at time $t$). Then,

- $S^t = S^{t-1}$.

- Drop from $S^t$ any $i$ such that $\sum_{j \in B_i} x_j^t < 1 - \epsilon$.

- **Iteratively:** as long as there exists a client $j \in C^t$ such that $j \notin \bigcup_{i \in S^t} \widehat{B_i^t}$ (uncovered by our current solution).

  - Add $j$ to $S^t$ (and $B_j$ is, as defined, of radius $r_j = D^t$).

  - Drop from $S^t$ any center/ball $i \in S^t, i \neq j$ such that

  $$d_{ij} \leq r_i + r_j + \delta \cdot \min\{r_i, r_j\}. \qquad (3.1)$$

**Analysis:** We will prove by induction on the steps of the algorithm that the algorithm maintains the following invariants.

(i) For each $j \in C^t$, we have $j \in \bigcup_{i \in S^t} \widehat{B_i^t}$.

(ii) For all $i_1, i_2 \in S^t$: $d_{i_1, i_2} \geq r_{i_1} + r_{i_2} + \delta \cdot \min\{r_{i_1}, r_{i_2}\}$. In particular the balls around the centers are disjoint.

(iii) For all $i \in S^t$, we have $\sum_{j \in B_i} x_j^t \geq 1 - \epsilon$.

(iv) For all $i \in S^t$, we have $\sum_{j \in B_i} x_j^{t_i} \geq 1$.

In addition, we use the following potential function at time $t \in [T]$, with parameter $\eta = 1/\log_2(\alpha - \delta - 1) = 1/\log_2(2 + \sqrt{2})$:

$$\Phi^t = \frac{1 + \eta \log \Delta}{\epsilon} \cdot \sum_{i \in S^t} \max\{0, 1 - \sum_{j \in B_i} x_j^t\}$$
$$- \eta \cdot \sum_{i \in S^t} \log_2\left(\frac{\Delta}{r_i}\right). \qquad (3.2)$$

Let $N_1^t$ be the total number of balls dropped due to the first step of the algorithm until time $t$. Let $N_2^t$ be the number of balls dropped until time $t$ due to an addition of a new ball in the second (iterative) step of the algorithm. We prove inductively that,

$$\Delta N_1^t + \Delta N_2^t + \Delta \Phi^t \le O\left(\frac{\log \Delta}{\epsilon}\right) \cdot \|x^t - x^{t-1}\|_1, \quad (3.3)$$

where $\Delta N_1^t, \Delta N_2^t$ and $\Delta \Phi^t$ is the change in the values of $N_1^t, N_2^t$ and $\Phi^t$ at time step $t$. In particular, we show that the RHS of the above equation as well as $\phi$ are finite. Therefore, the number of balls that are dropped due to an addition of a new ball at time step $t$ in the second iterative step, $\Delta N_2^t$, is finite and the algorithm terminates.

Before showing that the algorithm maintains these invariants, we show why these imply Theorem 3.1.

*Proof of Theorem 3.1.* By Invariant (i) and the definition of $\widehat{B_i^t}$ (recall, the radius of $\widehat{B_i^t}$ is at most $\alpha \cdot D^t \le \alpha \cdot \beta \cdot OPT^t$) the algorithm maintains at all times an $(\alpha \cdot \beta)$-approximate solution. By Invariant (ii) the balls are disjoint, and by Invariant (iii) we have for all $i \in S^t$ that $\sum_{j \in B_i} x_j^t \ge 1 - \epsilon$. Since the fractional solution satisfies $\sum_{i=1}^n x_i^t \le k'$ and $\epsilon \le 1/2$, the number of centers chosen by the algorithm is at most $\frac{k'}{1-\epsilon} \le (1 + 2\epsilon) \cdot k'$. Finally, by Inequality (3.3), $N_1^t + N_2^t + \Phi^t - \Phi^0 \le O(\log \Delta/\epsilon) \cdot R$. It is easy to verify that $\Phi^0 = 0$ and $\Phi^t \ge -\eta \cdot \sum_{i \in S^t} \log_2(\Delta/r_i) \ge -2k \log \Delta$. As the number of additions of centers is at most the total number of drops of centers plus the final number of centers (which is bounded by $k$), the total recourse is bounded as claimed. $\square$

**Remark 3.2.** *Our analysis has an additional (constant) additive term that depends on the final number of centers. However, a more careful inspection of our proof shows that the number drops is bounded by $O(\log \Delta/\epsilon)$ times the total decrease in the variables $x_i^t$. This fact along with the fact that the final number of centers plus $\eta \cdot \sum_{i \in S^t} \log_2(\Delta/r_i)$ is bounded by $O(\log \Delta/\epsilon)$ times the total increase in the variables can be used to avoid this additive term in the analysis.*

Next, we consider different events that can happen at time $t$ one-by-one and prove that all the invariants are maintained.

The invariants are clearly satisfied initially when there are no clients, the fractional solution is 0, and no centers were added. The following events may happen at time step $t$.

- The fractional solution changes from $x^{t-1}$ to $x^t$.

- Facility $i$ such that $\sum_{j \in B_i} x_j^t < (1 - \epsilon)$ is dropped in the first step.

- A new center/ball around $j$ is added, and then (possibly) other centers/balls are dropped in the second iterative step.

**The fractional solution changes:** By our induction hypothesis and Invariant (ii) we have that for all $i_1, i_2 \in S^t$: $d_{i_1, i_2} \ge r_{i_1} + r_{i_2} + \delta \cdot \min\{r_{i_1}, r_{i_2}\}$, and in particular the balls are disjoint. The only term that depends on the fractional solution is $(1+\eta \log \Delta)/\epsilon \cdot \sum_{i \in S^t} \max\{0, 1 - \sum_{j \in B_i} x_j^t\}$ in the potential function. Hence, Inequality (3.3) is maintained.

**Facility $i$ such that $\sum_{j \in B_i} x_j^t < 1 - \epsilon$ is dropped:** If this event happens, then $\Delta N_1^t = 1, \Delta N_2^t = 0$, and $\max\{0, 1 - \sum_{j \in B_i} x_j^t\} \ge \epsilon$. Thus, we have, $\Delta N_1^t + \Delta N_2^t + \Delta \Phi^t \le 1 - (1 + \eta \log_2 \Delta) + \eta \log_2\left(\frac{\Delta}{r_i}\right) \le 0$.

**A new ball around $j$ is added, and then (possibly) other balls are dropped:** In order to analyze this event we need the following lemma, which we prove shortly.

**Lemma 3.3.** *If $j \notin \bigcup_{i \in S^t} \widehat{B_i^t}$, and let $r_j = D^t$ be the radius of the new added ball $B_j$. Then, there exists at most one $i \in S^t$ such that $d_{ij} \le r_i + r_j + \delta \cdot \min\{r_i, r_j\}$. This ball (if exists) has radius $r_i > (\alpha - \delta - 1) \cdot D^t$. After adding this new ball and possibly dropping at most one ball all invariants (ii), (iii), (iv) are maintained.*

Given Lemma 3.3 we prove that the potential argument (Inequality (3.3)) is satisfied and the algorithm terminates. We observe that if indeed the main loop of the algorithm terminates after a finite number of steps, then by the main loop condition, Invariant (i) must be maintained.

Since by the LP constraints we have for the new added ball $\sum_{j \in B_i} x_j^t = \sum_{j \in B_i} x_j^{t_i} \ge 1$, the term $(1+\eta \log \Delta)/\epsilon \cdot \sum_{i \in S^t} \max\{0, 1 - \sum_{j \in B_i} x_j^t\}$ of the potential does not increases due to the addition of the new ball and possibly the drop of a single ball. Next, we have two cases.

**A ball of radius $r_i$ is added and no ball is dropped.** In this case $\Delta N_2^t = 0$ and the change in the LHS of Inequality (3.3) is at most $\Delta \Phi \le -\eta \log_2\left(\frac{\Delta}{r_j}\right) \le 0$. We note that by invariants (iv) for the new added ball $B_j$ we have $\sum_{i \in B_j} x_i^{t_j} \ge 1$. As by Invariant (ii) the balls are disjoint,

and $\sum_j x_j^t \leq k(1+\epsilon)$, this step can occur at most $k(1+\epsilon)$ times.

**A ball of radius $r_j$ is added, and a single ball $i$ with $r_i > (\alpha - \delta - 1) \cdot r_j$ is dropped.** The LHS of Inequality (3.3) increases by at most $\Delta N_2^t + \Delta\Phi \leq 1 - \eta \log_2\left(\frac{\Delta}{r_j}\right) + \eta\log_2\left(\frac{\Delta}{r_i}\right) = 1 - \eta\log_2\left(\frac{r_i}{r_j}\right) \leq 1 - \eta\log_2(\alpha - \delta - 1) \leq 0$, where the last inequality follows by the definition $\eta = 1/\log_2(\alpha - \delta - 1)$. As in each such step, the potential function decreases by at least 1, the potential function only decreases in the previous case, and $|\Phi^t| \leq O(k\log\Delta)$, this step can also happen at most $O(k\log\Delta)$ times. We conclude that the main loop of the algorithm must terminate after a finite number of steps.

*Proof of Lemma 3.3.* Since client $j$ is uncovered (i.e. $j \notin \bigcup_{i\in S^t} \widehat{B_i^t}$), we have $d_{ij} > \alpha \cdot D^t$ for all $i \in S^t$. The new added ball $B_j$ is of radius $r_j = D^t$. Thus, for any $i \in S^t$ such that $d_{ij} \leq r_i + r_j + \delta\min\{r_i, r_j\}$ we have,

$$r_i + (1+\delta)D^t = r_i + (1+\delta)r_j$$
$$\geq r_i + r_j + \delta \cdot \min\{r_i, r_j\} \geq d_{ij} > \alpha \cdot D^t.$$

Thus, for such a ball it must be that $r_i > (\alpha - \delta - 1) \cdot D^t$.

Assume for the sake of contradiction that there are two balls $B_{i_1}, B_{i_2}$ such that $d_{i_1,j} \leq r_{i_1} + r_j + \delta \cdot \min\{r_{i_1}, r_j\}$ and $d_{i_2,j} \leq r_{i_2} + r_j + \delta \cdot \min\{r_{i_2}, r_j\}$. Then

$$d_{i_1,i_2} \leq d_{i_1,j} + d_{i_2,j}$$
$$\leq r_{i_1} + r_{i_2} + 2r_j + \delta \cdot \min\{r_{i_1}, r_j\} + \delta \cdot \min\{r_{i_2}, r_j\}$$
$$= r_{i_1} + r_{i_2} + (2+2\delta) \cdot r_j = r_{i_1} + r_{i_2} + (2+2\delta) \cdot D^t$$
$$< r_{i_1} + r_{i_2} + \frac{2+2\delta}{\alpha - \delta - 1} \cdot \min\{r_{i_1}, r_{i_2}\}.$$

The first inequality follows by the triangle inequality. The second inequality follows by our assumption. The next two steps follow by our above observation $r_j = D^t$, and because $r_{i_1} > (\alpha - \delta - 1) \cdot D^t$ and $r_{i_2} > (\alpha - \delta - 1) \cdot D^t$. Finally, recalling our specific values of $\alpha = 3 + 2\sqrt{2}$ and $\delta = \sqrt{2}$, we get that this last line is at most $r_{i_1} + r_{i_2} + \delta \cdot \min\{r_{i_1}, r_{i_2}\}$, which contradicts the inductive hypothesis that Invariant (ii) holds before adding ball $B_j$.

After adding the new ball of radius $D^t$ and dropping at most one ball violating Invariant (ii), the new balls satisfy Invariant (ii). Finally, as $j \in C^t$ by the LP constraints $\sum_{j' \in B_j} x_{j'}^t = \sum_{j' \in B_j} x_{j'}^{t_i} \geq 1$ and therefore Invariant (iv) holds. □

## 3.2. A rounding algorithm for fully-dynamic facility location

We turn to our dynamic rounding scheme for facility location. Following Appendix A.2, for every $\epsilon \in (0, 1]$ we have

a fractional solution satisfying the guarantees

$$\sum_{t=1}^{T} \|x^t - x^{t-1}\|_1 \leq O\left(\frac{\log(\frac{|F|}{\epsilon})}{\epsilon}\right) \cdot \text{OPT}_{\text{REC}}^{\beta},$$
$$\sum_{i \in F} y_{ij}^t \geq 1 \qquad \forall j \in C^t, t \in [T],$$
$$y_{ij}^t \leq x_i^t \qquad \forall j \in C^t, t \in [T],$$
$$\sum_{i \in F} f_i x_i^t + \sum_{\substack{i \in F \\ j \in C^t}} d_{ij} y_{ij}^t \leq (1+\epsilon)\beta \cdot OPT^t \qquad \forall t \in [T].$$

The variable $x_i^t$ is the fraction to which facility at position $i \in F$ is open at time $t$ and $y_{ij}^t$ is the fraction by which client $j$ is served with facility $i$ at time $t$. The bound in Theorem 1.1 for the fully dynamic facility location problem is obtained by combining the guarantee on the fractional solution along with the following theorem that we prove, and choosing $\epsilon = 1$.

**Theorem 3.4** (Dynamic facility location rounding). *Let $x^t$ be a fractional solution to the dynamic facility location problem that maintains a solution that is a $\beta$-approximation at any time with total fractional recourse $R$. Then, there is an algorithm that maintains an **integral** solution that is $(\alpha \cdot \beta)$-approximate at any time, and its total recourse is $O(\log\Delta) \cdot R$, where $\alpha = 11$.*

**The Algorithm:** At any time $t$, for each $j \in C^t$, let $R_j^t = \sum_{i \in F} d_{ij} y_{ij}^t$ be the fractional connection cost of client $j$. The algorithm maintains at any time a set of active clients $A^t$ and a set of open facilities $S^t$. For each client $j \in A^t$ there is (exactly) a single associated facility $i_j \in S^t$. Whenever the algorithm adds or removes a client from $A^t$ it also adds/removes the associated facility from $S^t$. Let $t_i \leq t$ be the time a client $i$ (and the associated facility) were added to $A^t$ (correspondingly to $S^t$). We associate each $j \in A^t$ with a radius $r_j$ and a ball $B_j = \{i \in F \mid d_{ij} \leq r_j\}$. With foresight, set the parameters $\alpha = 11, \delta = 2, \gamma = 10/9, \eta = 1/\log((\alpha - \gamma(1+\delta))/2\gamma)$. At time $t$, the fractional solution is updated from $(x^{t-1}, y^{t-1})$ to $(x^t, y^t)$, and the algorithm is the following.

- $A^t \leftarrow A^{t-1}, S^t \leftarrow S^{t-1}$.

- Drop from $A^t$ (and correspondingly its associated facility $S^t$) any $j \in A^t$ such that $\sum_{i \in B_j} x_i^t < 1/\alpha$.

- **Iteratively:** as long as there exists $j \in C^t$ such that $d_{ij} > \alpha \cdot R_j^t$ for all $i \in S^t$.
    - Add the client $j$ to $A^t$, and set $r_j = \gamma \cdot R_j^t$.
    - For client $j$ add to $S^t$ a facility with minimal cost $f_i$ such that $d_{ij} \leq r_j$.
    - Drop from $A^t$ (and the corresponding facility in $S^t$) any $i \in A^t, i \neq j$ such that $d_{ij} \leq r_i + r_j + \delta \cdot \min\{r_i, r_j\}$.

**Analysis:** We prove inductively on the steps of the algorithm that the algorithm maintains the following invariants:

(i) Each $j \in C^t$, $\min_{i \in S^t} d_{ij} \leq \alpha \cdot R_j^t$.

(ii) For all $j_1, j_2 \in A^t$: $d_{j_1,j_2} \geq r_{j_1} + r_{j_2} + \delta \cdot \min\{r_{j_1}, r_{j_2}\}$. In particular the balls around the clients in $A^t$ are disjoint.

(iii) For all $j \in A^t$, we have $\sum_{i \in B_j} x_i^t \geq 1/\alpha = 1/11$.

(iv) For all $j \in A^t$, we have $\sum_{i \in B_j} x_i^{t_j} \geq 1 - 1/\gamma = 1/10$.

In addition, we use the following potential function at time $t \in [T]$.

$$\Phi^t = \begin{pmatrix} \left(1 - \dfrac{1}{\gamma} - \dfrac{1}{\alpha}\right)^{-1} \cdot (1 + \eta \cdot \log_2(\Delta)) \\ \cdot \displaystyle\sum_{i \in A^t} \max\left\{0, 1 - \dfrac{1}{\gamma} - \displaystyle\sum_{j \in B_i} x_j^t\right\} \\ - \displaystyle\sum_{i \in A^t} \eta \cdot \log_2\left(\dfrac{\Delta}{r_i}\right). \end{pmatrix}$$

(3.4)

Let $N_1^t$ be the total number of balls dropped due to the first step of the algorithm until time $t$. Let $N_2^t$ be the number of balls dropped due to an addition of a new ball in the second (iterative) step of the algorithm. We prove inductively that,

$$\Delta N_1^t + \Delta N_2^t + \Delta \Phi^t \leq O(\log \Delta) \cdot \|x^t - x^{t-1}\|_1, \quad (3.5)$$

where $\Delta N_1^t, \Delta N_2^t$ and $\Delta \Phi^t$ is the change in the values of $N_1^t, N_2^t$ and $\Phi^t$ at time step $t$. In particular, we show that the RHS of the above equation as well as $\phi$ are finite. Therefore, the number of balls that are dropped due to an addition of a new ball at time step $t$ in the second iterative step, $\Delta N_2^t$, is finite and the algorithm terminates. We show first that these invariants imply Theorem 3.4.

*Proof of Theorem 3.4.* By Invariant (i), the total service cost is at most $\alpha \cdot \sum_{j \in C^t} R_j^t$.

Since $i_j$, the facility associated with client $j \in A^t$, is of minimal cost in $B_j$, we have $f_{i_j} \leq \sum_{i \in B_j} f_i \cdot x_i^t/(\sum_{i \in B_j} x_i^t) \leq \alpha \cdot \sum_{i \in B_j} f_i \cdot x_i^t$, where the final inequality is by Invariant (iii) that we maintain for all $j \in A^t$, $\sum_{i \in B_j} x_i^t \geq 1/\alpha$. By Invariant (ii) the balls $B_j$ are disjoint, and hence the total cost of opening facilities is at most $\alpha \cdot \sum_{i \in F} f_i x_i^t$. Therefore, our solution is an $(\alpha \cdot \beta)$-approximation.

Finally, by Inequality (3.5) $N_1^t + N_2^t + \Phi^t - \Phi^0 \leq O(\log \Delta) \cdot \sum_{t=1}^{T} \|x^t - x^{t-1}\|_1$. It is easy to verify that $\Phi^0 = 0$ and $\Phi^t \geq -(1 - 1/\gamma - 1/\alpha)^{-1} \cdot (1 + \eta \cdot \log_2(\Delta)) \cdot \sum_{i \in A^t} \sum_{j \in B_i} x_i^t \geq -C \log \Delta \sum_{i \in F} x_i^t$, where $C$ is some constant. Thus, the total recourse is bounded. $\square$

*Remark 3.5. Our analysis has an additional (constant) additive term that depends on the final fractional solution. However, a more careful inspection of our proof shows that the number balls dropped is bounded by $O(\log \Delta)$ times the total decrease in the variables $x_i^t$. This fact along with the fact that the final number of facilities plus $O(\log \Delta) \cdot \sum_{i \in F} x_i^t$ is bounded by $O(\log \Delta)$ times the total increase in the variables $x_i^t$ can be used to avoid this additive term in the analysis.*

Next, we consider different events that can happen at time step $t$ one-by-one and prove that all the invariants are maintained. The invariants are clearly satisfied initially when there are no clients, the fractional solution is $0$, and no facilities were added. The following events may happen.

- The fractional solution changes.

- Client $j$ such that $\sum_{i \in B_j} x_i^t < 1/\alpha$ is dropped.

- A new ball around client $j$ is added, and then (possibly) other clients/balls are dropped.

**The fractional solution changes:** By our induction and Invariant (ii) we have that for all $i_1, i_2 \in S^t$: $d_{i_1,i_2} \geq r_{i_1} + r_{i_2} + \delta \cdot \min\{r_{i_1}, r_{i_2}\}$, and in particular the balls are disjoint. Only the term $(1 - 1/\gamma - 1/\alpha)^{-1} \cdot (1 + \eta \cdot \log_2(\Delta)) \cdot \sum_{i \in A^t} \max\left\{0, (\gamma - 1)/\gamma - \sum_{j \in B_i} x_j^t\right\}$ depends on the fractional solution and Inequality (3.5) is maintained.

**Client $i$ (and its corresponding facility) such that $\sum_{j \in B_i} x_j^t < \frac{1}{\alpha}$ is dropped:** If this event happens, then $\Delta N_1^t = 1, \Delta N_2^t = 0$, and $\max\{0, 1 - 1/\gamma - \sum_{j \in B_i} x_j^t\} \geq 1 - 1/\gamma - 1/\alpha$. Thus, we have,

$$\Delta N_1^t + \Delta N_2^t + \Delta \Phi^t$$
$$\leq 1 - \frac{1 - 1/\gamma - 1/\alpha}{1 - 1/\gamma - 1/\alpha}(1 + \eta \cdot \log_2(\Delta)) + \eta \cdot \log_2\left(\frac{\Delta}{r_i}\right)$$
$$\leq 0.$$

**A new ball around $j$ is added, and then (possibly) other balls are dropped:** In order to analyze this event we need the following lemma, which we prove shortly.

**Lemma 3.6.** *Suppose the algorithm adds a client $j$ such that $d_{ij} > \alpha \cdot R_j^t$ for all $i \in S^t$, and let $r_j = \gamma \cdot R_j^t$ be the radius of the new added ball $B_j$ around $j$. Then, there exists at most one client $i \in A^t$ such that $d_{ij} \leq r_i + r_j + \delta \cdot \min\{r_i, r_j\}$, and if such client exists then $r_i > 2^{1/\eta} \cdot r_j$. After adding this new client/ball and possibly dropping at most one client/ball all invariants (ii), (iii), (iv) are maintained.*

Given Lemma 3.6, we only need to prove that the potential argument (Inequality (3.5)) is satisfied. Since by

the LP constraints and markov inequality we have for the new added ball $\sum_{i \in B_j} x_i^t = \sum_{i \in B_j} x_i^{t_j} \geq 1 - 1/\gamma$, then the term $(1 - 1/\gamma - 1/\alpha)^{-1} \cdot (1 + \eta \cdot \log_2(\Delta)) \cdot \sum_{i \in A^t} \max\left\{0, (\gamma - 1)/\gamma - \sum_{j \in B_i} x_j^t\right\}$ of the potential does not increases due to the addition of the new ball and possibly the drop of a single ball. Next, we have two cases.

**A ball of radius $r_j$ is added and no ball is dropped.** In this case $\Delta N_2^t = 0$ and the change in the LHS of Inequality (3.5) is at most $\Delta\Phi \leq -\eta \cdot \log_2(\Delta/r_j) \leq 0$.

**A ball of radius $r_j$ is added a single ball $i$ with $r_i > 2^{1/\eta} \cdot r_j$ is dropped.** The LHS of Inequality (3.5) increases by at most $\Delta N_2^t + \Delta\Phi \leq 1 - \eta \cdot \log_2(\Delta/r_j) + \eta \cdot \log_2(\Delta/r_i) = 1 - \eta \cdot \log_2(r_i/r_j) \leq 1 - \eta \cdot \log_2(2^{1/\eta}) \leq 0$.

*Proof of Lemma 3.6.* For the client $j$, we have $d_{ij} > \alpha \cdot R_j^t$ for all $i \in S^t$. The new added ball $B_j$ is of radius $r_j = \gamma \cdot R_j^t$. Let $i \in A^t$ be any client such that $d_{ij} \leq r_i + r_j + \delta \cdot \min\{r_i, r_j\}$, and let $i'$ be the associated facility of $i$. We have $2r_i + (1 + \delta)r_j \geq r_i + r_i + r_j + \delta \cdot \min\{r_i, r_j\} \geq d_{ii'} + d_{ij} \geq d_{i'j} > \alpha \cdot R_j^t = \frac{\alpha}{\gamma} r_j$, where the second inequality follows by the guarantee that facility $i'$ is at distance at most $r_i$ from client $i$, and the guarantee on $d_{ij}$. The third Inequality is by the triangle inequality. The last inequality holds because $d_{i'j} > \alpha \cdot R_j^t$.

Thus, for such a ball it must be that $r_i > \frac{(\frac{\alpha}{\gamma} - (1+\delta))/2 \cdot r_j}{} = 2^{1/\eta} \cdot r_j$ (recall that $\eta = 1/\log((\alpha - \gamma(1+\delta))/2\gamma)$). Next, assume to the contrary that there are two balls $B_{i_1}, B_{i_2}$ such that $d_{i_1,j} \leq r_{i_1} + r_j + \delta \cdot \min\{r_{i_1}, r_j\}$ and $d_{i_2,j} \leq r_{i_2} + r_j + \delta \cdot \min\{r_{i_2}, r_j\}$. Then

$$d_{i_1,i_2} \leq d_{i_1,j} + d_{i_2,j}$$
$$\leq r_{i_1} + r_{i_2} + 2r_j + \delta \cdot \min\{r_{i_1}, r_j\} + \delta \cdot \min\{r_{i_2}, r_j\}$$
$$\leq r_{i_1} + r_{i_2} + (2 + 2\delta) \cdot r_j$$
$$\leq r_{i_1} + r_{i_2} + \frac{4 + 4\delta}{\alpha/\gamma - (1 + \delta)} \cdot \min\{r_{i_1}, r_{i_2}\}.$$

The first inequality follows by the triangle inequality. The second inequality follows by our assumption. The third and fourth inequality follow by our above observation that $r_{i_1} > 2^{1/\eta} \cdot r_j$ and $r_{i_2} > 2^{1/\eta} \cdot r_j$. Finally, this is strictly less than $r_{i_1} + r_{i_2} + \delta \cdot \min\{r_{i_1}, r_{i_2}\}$ by our specific setting of the parameters. This contradicts the induction hypothesis that Invariant (ii) held before ball $B_j$ was added.

After adding the new ball of radius $r_j$, and dropping at most one ball that violates Invariant (ii), the new balls satisfy Invariant (ii). Finally, because $j \in A^t$ by the LP constraints, by Markov's Inequality (since $r_j = \gamma \cdot R_j^{t_j}$) $\sum_{i \in B_j} x_i^t = \sum_{i \in B_j} x_i^{t_j} \geq 1 - 1/\gamma$ and therefore Invariant (iv) is maintained. $\square$

### 3.3. A rounding algorithm for fully dynamic $k$-median

With the setup from previous sections, our algorithm for $k$-median is very simple to describe and to analyze. Given a fractional solution for the problem formulation in Appendix A.3, we run the rounding algorithm from Section 3.2, with facility costs $f_i = 0$ for all $i \in M$. The bound in Theorem 1.1 for the fully dynamic $k$-median problem is obtained by combining the guarantee on the fractional solution along with the following theorem that we prove.

**Theorem 3.7** (Dynamic $k$-median rounding). *Let $x^t$ be a fractional solution to the dynamic $k$-median problem that maintains a solution that is a $\beta$-approximation at any time, uses at most $k'$ centers, and its total fractional recourse is $R$. Then there is an algorithm that maintains an **integral** solution with at most $\alpha \cdot k'$ centers that is $(\alpha \cdot \beta)$-approximate at any time, and its total recourse is $O(\log(\Delta)) \cdot R$, where $\alpha = 11$.*

*Proof.* The approximation and recourse guarantees are immediate from Theorem 3.4, and it remains to show that the algorithm never holds more than $O(k)$ centers.

Inspecting the proof of Theorem 3.4, by Invariant (ii) the balls $B_j$ are disjoint, and by Invariant (iii) we have for all $i \in F^t$ that $\sum_{j \in B_i} x_j^t \geq 1/\alpha$. Since the fractional solution satisfies $\sum_{i=1}^n x_i^t \leq k'$, the number of centers chosen by the algorithm is at most $\alpha \cdot k'$. $\square$

## 4. Experiments

We evaluate our three clustering algorithms experimentally on UCI Machine Learning repository datasets: Glass Identification (German, 1987), Wine Quality (Cortez et al., 2009), and Airfoil Self-Noise (Brooks et al., 2014). Due to space considerations, we defer details of our experimental setup and the plots of our experiments to Appendix D. We observe that our algorithm indeed maintains an objective value within the predicted bound, and in fact significantly outperforms the worst-case bound predicted by the theorem. In the case of facility location and $k$-median, our objective value is almost the same as the best possible objective value. On all of our data sets, our algorithm's recourse seems to be $\ll T$, and even seems bounded by a constant in many cases. This justifies our focus on *competitive* recourse, as opposed to absolute recourse. Interestingly, in many cases (especially for $k$-center), our online algorithm's recourse is *lower* than the offline fractional optimum recourse. Note that this is possible because our algorithms is using resource augmentation, while the fractional optimum is not. Finally, we observe that in our experiments for both $k$-center and $k$-median, even though our algorithm is allowed to open $(1 + \epsilon)$ centers, they tend to use no more than $k$ centers.

# 5. Conclusion

In this work we initiate the study of consistent clustering with competitive recourse guarantees. We show how to maintain $O(\beta)$ approximations for $k$-center, facility location and $k$-median with $O(\log|F|\log\Delta)\cdot\mathrm{OPT}_{\mathrm{REC}}^{\beta}$ recourse (using $O(k)$ centers for $k$-center and $k$-median), and showed that any such algorithm must incur recourse of at least $\Omega(\min\{\log|F|,\log\Delta)\})\cdot\mathrm{OPT}_{\mathrm{REC}}^{\beta}$.

There are plenty of natural open questions. The most obvious is to close the gap between upper and lower bounds. Another interesting question is to understand whether using $(1+\epsilon)\cdot k$ centers (or $O(k)$ centers) is necessary, or whether one can do with exactly $k$ centers. A weaker interim goal is to give an algorithm for $k$-median which only opens $(1+\epsilon)\cdot k$ centers, without suffering a $1/\epsilon$ factor in the approximation. We note even offline rounding algorithms for $k$-median are fairly involved (Charikar et al., 2002), and historically came after simpler bicriteria rounding algorithms (Lin & Vitter, 1992). It would also be beneficial to better understand the relationship between algorithms with absolute resource guarantees and algorithms with competitive recourse guarantees. It would be interesting to benchmark our algorithms on real world datasets, and in particular to evaluate how they compare to existing algorithms with absolute recourse guarantees. Finally, it would be interesting to explore competitive recourse algorithms for other clustering objectives in the fully dynamic model.

## Disclosure of Funding

The work of Niv Buchbinder is supported in part by the Israel Science Foundation (ISF) grant no. 3001/24, and the United States - Israel Binational Science Foundation (BSF) grant no. 2022418.

## Impact Statement

This paper presents work whose goal is to advance the field of Machine Learning. There are many potential societal consequences of our work, none which we feel must be specifically highlighted here.

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

# A. Formulations as positive body chasing

The fractional versions of our clustering problems is captured by a framework referred to as *Positive Body Chasing* suggested recently in (Bhattacharya et al., 2023). In the chasing positive bodies problem, we are given a sequence of bodies $K_t = \{x^t \in \mathbb{R}_+^n \mid C^t x^t \geq 1, P^t x^t \leq 1\}$ revealed online, where $C^t$ and $P^t$ are matrices with non-negative entries. The goal is to (approximately) maintain a point $x^t \in K_t$ such that the total $\ell_1$-movement, $\sum_t \|x^t - x^{t-1}\|_1$, is minimized where $x^0 = 0$. More generally, given weight $w \in \mathbb{R}_+^n$, we want to minimize the weighted $\ell_1$-movement, $\sum_t w_i \sum_{i=1}^n |x_i^t - x_i^{t-1}|$. We sometimes refer to this $\ell_1$ movement as *fractional recourse*. In (Bhattacharya et al., 2023), the authors designed an online algorithm for this problem proving the following theorem.

**Theorem A.1** ((Bhattacharya et al., 2023)). *For any $\epsilon \in (0, 1]$, there is an $O\left(\frac{1}{\epsilon} \log\left(\frac{d}{\epsilon}\right)\right)$-competitive algorithm for chasing positive bodies in $\ell_1$ such that $x^t \in K_t^{1+\epsilon} = \{x^t \in R_+^n \mid C^t x^t \geq 1, P^t x^t \leq 1 + \epsilon\}$ at time t, and d is the maximal number of non-negative coefficients in a covering constraint.*

We next list the dynamic clustering problems that we study, formulate each problem in the framework of Theorem A.1, and derive the guarantees on the fractional solution that is produced in an online fashion.

## A.1. Fractional dynamic $k$-center

Let $x_i^t$ be an indicator for the opening of a center at position $i \in M$ at time $t$. For each $j \in C^t$ let $B_j^t = \{i \in M \mid d_{ij} \leq \min\{\beta \cdot OPT^t, \Delta\}\}$ be the set of points that are within distance at most $\beta \cdot OPT^t$ from client $j$. The set $B_j^t$

is the set of points that can serve the client at time $t$ in a $\beta$-approximate solution. Given these variables, the following is a formulation of the offline fully dynamic $k$-center problem.

$$
\left\{ \min \sum_{t=1}^{T} \|x^t - x^{t-1}\|_1 \;\middle|\; 
\begin{array}{ll}
\sum_{i \in B_j^t} x_i^t \geq 1, & \\
& \forall j \in C^t, t \in [T] \\
\sum_{i=1}^{n} x_i^t \leq k, & \\
& \forall t \in [T] \\
x_i^t \geq 0, & \\
& \forall i \in M, t \in [T]
\end{array}
\right\}
$$

Let $x^*$ be the optimal fractional solution, and let $\mathrm{OPT}_{\mathrm{REC}}^{\beta} = \sum_{t=1}^{T} \|x^{*t} - x^{*(t-1)}\|_1$. As the set of constraints at time $t$ is a covering/packing formulation, the fractional fully dynamic $k$-center problem is captured by the positive body chasing problem. Hence, by Theorem A.1 we get the following guarantee. For any $\epsilon \in (0, 1]$, there is an online algorithm that produces a fractional solution $x \geq 0$ to the fully dynamic $k$-center problem such that,

$$
\begin{aligned}
\sum_{t=1}^{T} \|x^t - x^{t-1}\|_1 &\leq O\left(\frac{\log(n/\epsilon)}{\epsilon}\right) \cdot \mathrm{OPT}_{\mathrm{REC}}^{\beta}, \\
\sum_{i \in B_j^t} x_i^t &\geq 1 \qquad \forall j \in C^t, t \in [T], \\
\sum_{i=1}^{n} x_i^t &\leq (1+\epsilon)k \qquad \forall t \in [T].
\end{aligned}
$$

## A.2. Fractional facility location

Let $x_i^t$ be an indicator for the opening of a facility at position $i \in F$ at time $t$. Let $y_{ij}^t$ be an indicator for serving client $j$ with facility $i$ at time $t$. Given these variables, the following is a formulation of the offline fully dynamic facility location problem.

$$
\left\{ \min \sum_{t=1}^{T} \|x^t - x^{t-1}\|_1 \;\middle|\;
\begin{array}{l}
\sum_{i \in F} y_{ij}^t \geq 1, \\
\qquad \forall j \in C^t, t \in [T] \\
y_{ij}^t \leq x_i^t, \\
\qquad \forall j \in C^t, i \in F, t \in [T] \\
\sum_{i \in F} f_i x_i^t + \sum_{i \in F, j \in C^t} d_{ij} y_{ij}^t \\
\qquad \leq \beta \cdot OPT^t, \quad \forall t \in [T] \\
x_i^t, y_{ij}^t \geq 0, \\
\qquad \forall i \in F, j \in C^t, t \in [T]
\end{array}
\right\}
$$

Although the above formulation is not a covering-packing LP due to the constraint $y_{ij}^t \leq x_i^t$, we show in Appendix B that it is possible to work with an equivalent formulation that is a covering-packing. This means that the fractional fully dynamic facility location problem is captured by the positive body chasing problem. In particular, we compete with an optimal solution to the modified formulation which

is at most the optimal solution to the original formulation, and can transform the resulting fractional solution online to a solution to the original LP at no additional cost. Note also that we do not pay recourse for changing the variables $y_{ij}$. This is captured by giving these variables weight zero in the weighted $\ell_1$-norm of the positive body chasing problem. Let $x^*, y^*$ be the optimal fractional solution, and let $\mathrm{OPT}_{\mathrm{REC}}^{\beta} = \sum_{t=1}^{T} \|x^{*t} - x^{*(t-1)}\|_1$. Hence, by Theorem A.1 we get the following guarantee. For any $\epsilon \in (0, 1]$, there is an online algorithm that produces a fractional solution $(x, y) \geq 0$ to the fully dynamic facility location problem such that,

$$
\begin{aligned}
\sum_{t=1}^{T} \|x^t - x^{t-1}\|_1 &\leq O\left(\frac{\log(|F|/\epsilon)}{\epsilon}\right) \cdot \mathrm{OPT}_{\mathrm{REC}}^{\beta}, \\
\sum_{i \in F} y_{ij}^t &\geq 1 \qquad \forall j \in C^t, t \in [T], \\
y_{ij}^t &\leq x_i^t \qquad \forall i \in F, j \in C^t, t \in [T], \\
\sum_{i \in F} f_i x_i^t + \sum_{i \in F, j \in C^t} d_{ij} y_{ij}^t &\leq (1+\epsilon) \cdot \beta \cdot OPT^t \\
& \qquad \forall t \in [T].
\end{aligned}
$$

## A.3. Fractional $k$-median

Let $x_i^t$ be an indicator for the opening of a center at position $i \in M$ at time $t$. Let $y_{ij}^t$ be an indicator for serving client $j$ with center $i$ at time $t$. Given these variables, the following is a formulation of the offline fully dynamic $k$-median problem.

$$
\left\{ \min \sum_{t=1}^{T} \|x^t - x^{t-1}\|_1 \;\middle|\;
\begin{array}{l}
\sum_{i \in M} y_{ij}^t \geq 1, \\
\qquad \forall j \in C^t, t \in [T] \\
y_{ij}^t \leq x_i^t, \\
\qquad \forall j \in C^t, i \in M, t \in [T] \\
\sum_{i \in M} x_i^t \leq k, \\
\qquad \forall t \in [T] \\
\sum_{\substack{i \in M \\ j \in C^t}} d_{ij} y_{ij}^t \leq \beta \cdot OPT^t, \\
\qquad \forall t \in [T]
\end{array}
\right\}
$$

As in the case of facility location, we can transform this into a covering-packing formulation (See Appendix B). Once again by Theorem A.1, for any $\epsilon \in (0, 1]$, there is an online algorithm that produces a fractional solution $(x, y) \geq 0$ to

the fully $k$-median problem such that:

$$
\sum_{t=1}^{T} \|x^t - x^{t-1}\|_1 \le O\left(\frac{\log(\frac{n}{\epsilon})}{\epsilon}\right) \cdot \mathrm{OPT}_{\mathrm{REC}}^{\beta},
$$

$$
\sum_{i \in M} y_{ij}^t \ge 1 \qquad\qquad \forall j \in C^t, t \in [T],
$$

$$
y_{ij}^t \le x_i^t \qquad\qquad \forall i \in M, j \in C^t, t \in [T],
$$

$$
\sum_{i \in M} x_i^t \le (1+\epsilon) \cdot k \qquad\qquad \forall t \in [T],
$$

$$
\sum_{\substack{i \in M \\ j \in C^t}} d_{ij} y_{ij}^t \le (1+\epsilon) \cdot \beta \cdot OPT^t \qquad\qquad \forall t \in [T].
$$

## B. A covering-packing LP formulation for the facility location and the $k$-median Problems

We show here that the facility location problem can be captured by a covering-packing formulation. The arguments for the $k$-median problem are similar.

Recall, the formulation in Section A.2. Let $x_i^t$ be an indicator for the opening of a facility at position $i \in F$ at time $t$. Let $y_{ij}^t$ be an indicator for serving client $j$ with facility $i$ at time $t$. Let $OPT^t$ be the optimal facilitiy location objective at time $t$. Given these variables, the following is a formulation of the offline fully dynamic facility location problem.

$$
\min \sum_{t=1}^{T} \|x^t - x^{t-1}\|_1,
$$

$$
\sum_{i \in F} y_{ij}^t \ge 1 \qquad\qquad \forall j \in C^t, t \in [T],
$$

$$
y_{ij}^t \le x_i^t \qquad\qquad \forall j \in C^t, t \in [T],
$$

$$
\sum_{i \in F} f_i x_i^t + \sum_{i \in F, j \in C^t} d_{ij} y_{ij}^t \le \beta \cdot OPT^t \qquad \forall t \in [T],
$$

$$
x_i^t, y_{ij}^t \ge 0 \qquad\qquad \forall i \in F, j \in C^t, t \in [T].
$$

The above formulation is not a covering-packing formulation. However, it is not hard to transform it into a covering packing formulation by removing the constraint $y_{ij}^t \le x_i^t$ and introducing in addition to the constraint $\sum_{i \in F} y_{ij}^t \ge 1$, the following exponential number of constraints:

$$
\sum_{i \in F'} y_{ij}^t + \sum_{i \in F \setminus F'} x_i^t \ge 1 \qquad \forall j \in C^t, F' \subseteq F, t \in [T].
$$

$$\text{(B.1)}$$

We observe the following.

**Lemma B.1.** *The modified formulation is separable in polynomial time. Moreover,*

- *Any solution to the original LP is feasible to the new formulation.*

- *A solution to the modified formulation can be transformed online into a feasible solution to the original formulation with the same recourse.*

*Proof.* Let $(x, y)$ be a solution to the original formulation. We need to prove that it satisfies all the new constraints of the form (B.1). This is true since,

$$
\sum_{i \in F'} y_{ij}^t + \sum_{i \in F \setminus F'} x_i^t \ge \sum_{i \in F} y_{ij}^t \ge 1,
$$

where the first inequality holds since $y_{ij}^t \le x_i^t$, and the second inequality holds by the original constraint that $\sum_{i \in F} y_{ij}^t \ge 1$.

On the other direction, if $(x, y)$ is a solution to the modified LP, then setting $y_{ij}'^t = \min\{y_{ij}^t, x_i^t\}$ satisfies the constraint $y_{ij}^t \le x_i^t$. Next, if $\sum_{i \in F} y_{ij}'^t = \sum_{i \in F} \min\{y_{ij}^t, x_i^t\} < 1$, it means that for $F' = \{i \in F \mid x_i^t \ge y_{ij}^t\}$,

$$
\sum_{i \in F'} y_{ij}^t + \sum_{i \in F \setminus F'} x_i^t = \sum_{i \in F} \min\{y_{ij}^t, x_i^t\} < 1.
$$

This contradicts the feasibility of the solution $x, y$ to the modified LP. Hence, the solution $y_{ij}'^t$ is feasible to the original LP. Moreover, since we didn't modify the variable $x_i^t$, it has the same recourse.

Finally, given a solution $x, y$ to the modified LP checking for each client $j$, whether the constraint with $F' = \{i \in F \mid x_i^t \ge y_{ij}^t\}$ satisfies $\sum_{i \in F'} y_{ij}^t + \sum_{i \in F \setminus F'} x_i^t = \sum_{i \in F} \min\{y_{ij}^t, x_i^t\} \ge 1$ suffices for the feasibility of the LP, and otherwise we get a violated constraint. $\square$

## C. Lower bounds

In this section we prove the following lower bounds for any randomized dynamic clustering algorithm. We draw ideas from (Fotakis, 2008).

**Theorem 1.2** (Lower Bound)**.** *Suppose there is a randomized algorithm that given a parameter $\beta \ge 1$ as an input maintains an $(\alpha \cdot \beta)$-approximation for one of fully-dynamic facility location, $k$-center, or $k$-median. Then this algorithm has recourse at least $\Omega(\min\{\log |F|, \log_\alpha \Delta\} \cdot \mathrm{OPT}_{\mathrm{REC}}^{\beta}$. For $k$-center and $k$-median, the lower bound holds even if the algorithm is allowed to open $O(k)$ facilities.*

*Proof.* The lower bounds share common ideas, but the exact metric, the adversarial sequence, and the cost analysis are different for each clustering problem. Therefore, we prove the lower bounds separately for each problem. For all these problems, we prove a lower bound for an online fractional solution that may generate a fractional solution to the linear formulation. Since the marginal probabilities of any randomized algorithm induce a fractional solution with

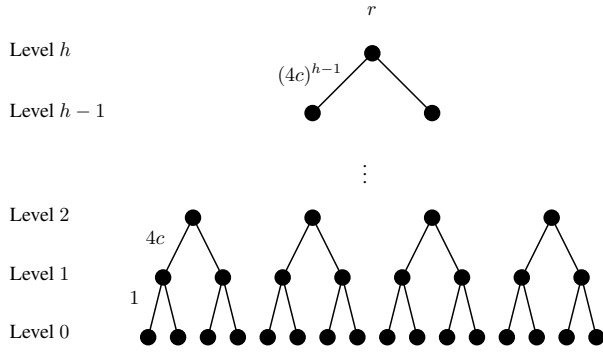

Level $h$

$(4c)^{h-1}$

Level $h-1$

⋮

Level 2

$4c$

Level 1

$1$

Level 0

*Figure 1.* Illustration of the HST-metric $H$.

at most the same cost, such a bound immediately translates to a bound on any randomized algorithm.

We will reuse the same base metric for our bounds. Let $H$ be a uniform binary $(4c)$-HST with $n$ leaves and height $h = \log_2 n$. The leaves of the tree are at level 0, and the root of the tree, $r$, is at level $h$. The edges that connect a node at level $i-1$ to a node at level $i$ (for $i = 1, \ldots, h-1$) have cost $(4c)^{i-1}$. For a non-leaf node $w$, let $w_R$ and $w_L$ be the right and left child of $w$ respectively.

**A lower bound for the fully-Dynamic $k$-center problem.** Assume that there exist a fractional algorithm that maintains at all times at most $k$ centers with an approximation ratio strictly less than $c$ with respect to the optimal solution at time $t$ (we later extend this bound to a scenario in which the algorithm is allowed to pen $b \cdot k$ facilities). We prove that such an algorithm has recourse competitiveness $\Omega(\min\{\log n, \log_c \Delta\})$, and our bounds hold even when $k = 1$. Once again we reuse the $(4c)$-HST $H$ defined above. This time, the metric space, as well as the possible locations for the clients and facilities is defined only by the $n$ leaves of the tree and the distances induced by the binary HST.

The adversarial sequence again happens in phases, and each phase is divided into $h$ time steps. At time 0, initialize the node $r^0$ to be $r$, the global root of the metric $H$. For every time $t = 0, \ldots, h$, the adversary creates two clients at $u_L^t$ and $u_R^t$, where these are the leftmost and rightmost leaves descendant of $r^t$ respectively. The adversary then removes any clients requested in the last time step. Let $x_L^t$ and $x_R^t$ be the total fraction of facilities that the algorithm opens in the subtrees of $r_L^t$ and $r_R^t$ (the two children of the node $r^t$). If $x_L^t \geq x_R^t$ then set the node $r^{t+1} := r_R^t$, otherwise set $r^{t+1} := r_L^t$. Finally, at iteration $t = h$ there is a single client located at a single leaf $v$. After time $h$, the adversary removes all clients, and we may repeat this phase/sequence starting from the root.

We observe that in every phase, opening a center on the last node $v$ of the sequence gives the lowest possible value for

the $k$-center objective for every time $t$ in the phase simultaneously. The cost of this solution at any round $t = 0, \ldots, h-1$ is

$$\text{OPT}^t = 2 \cdot \sum_{k=0}^{h-1-t} (4c)^k \leq 2 \cdot (4c)^{h-1-t}.$$

At time $t = h$, we have $\text{OPT}^h = 0$. The recourse of this solution is 1 per phase.

On the other hand, we also observe that if at time $t = 1, \ldots, h-1$ the algorithm has $x_L^t + x_R^t \leq 1/2$, then the algorithm's solution has cost at least $c \cdot \text{OPT}^t$. To see this, note that the two clients at $u_L^t$ and $u_R^t$ must be matched fractionally to extent at least $1/2$ to a center outside the subtree rooted at $r^t$. The distance to this center is at least twice the cost of the edge between $r^t$ and its parent $r^{t-1}$, which is at least

$$\frac{1}{2} \cdot 2 \cdot (4c)^{h-t} > c \cdot 2(4c)^{h-t-1} \geq c \cdot \text{OPT}^t.$$

Therefore, any algorithm maintaining an approximation better than $c$ with respect to $\text{OPT}^t$ must have at each time $t = 1, \ldots, h-1$ at least a $1/2$ fraction of a center open in the subtree of $r^t$. Since the algorithm can open at most one center at any given time, and by construction of the adversarial sequence, at every time $t = 0, \ldots, h-1$, the algorithm has more fractional mass in the child of $r^t$ that is not $r^{t+1}$, the algorithm must open at least $1/4$ of a center to maintain a $c$-approximation. Hence the algorithm must open a total of $\Omega(h)$ centers over the course of the phase, where we observe that size of the metric is $n = 2^h$ and the aspect ratio is $\Delta = O((4c)^h)$. This sequence may be repeated.

Finally, suppose that the algorithm is allowed to maintain a total mass of $b \cdot k = b$ facilities at each time step. Then, at the beginning of each phase, there exists a subtree of height $h' = \Omega(h - \log_2 b)$ with total initial mass of at most $1/4$ (the subtree with minimal mass among the $4b$ disjoint subtrees of height $h - \log_2(4b)$). The adversary can restrict its sequence to this subtree and the algorithm, again, must open $1/4$ of a facility in each level of the tree, paying a total opening cost of $\Omega(h - \log_2 b)$ during the phase. Overall, the lower bound on the recourse competitiveness is $\Omega(\min\{\log n, \log_c \Delta\} - \log b)$. $\qquad \square$

**A lower bound for the fully-dynamic facility location problem.** Assume that there exists a fractional algorithm that maintains at all times an approximation ratio strictly less than $c$ with respect to the optimal solution at time $t$. We prove that such an algorithm has recourse competitiveness $\Omega(\min\{\log|F|, \log_c \Delta\}$.

The underlying metric is the $(4c)$-HST $H$ defined above. The set of possible facility locations, $F$, is the set of $n$ leaves of the tree, but we allow clients to arrive at any internal node.

Finally, the cost of opening a facility is $f = (4c)^{h-1}$ (in other words, the cost is uniform).

The request sequence is in phases, each of which is divided into $h$ time steps. At time 0 there is a single client located at the root. Henceforth, for all time $t = 1, \ldots, h$, let $u^t$ be the node at which clients appeared at time $t-1$, and let $x_L^t$ and $x_R^t$ be the fractional mass of facilities that the algorithm allocates in the subtrees of $u_L^t$ and $u_R^t$ respectively. If $x_L^t \geq x_R^t$ then the adversary generates $(4c)^t$ clients at $u_R^t$; if $x_L^t < x_R^t$ then the adversary generates $(4c)^t$ clients at $u_L^t$ instead. The adversary then removes any client requests at $u^t$. Eventually, at time $h$, a set of $(4c)^h$ clients arrive at some leaf $v$. After time $h$, all clients leave, and we may repeat this phase/sequence starting from the root.

Note that in every phase, opening a single facility on the last leaf $v$ of the sequence gives the lowest possible value for the facility location objective at every time $t$ in this phase. This strategy uses total recourse of 1 per phase (opening a single facility at the beginning of the phase and closing it at the end of the phase). The cost of this solution at any round $t = 0, \ldots, h-1$ is:

$$\text{OPT}^t = f + (4c)^t \cdot \sum_{k=0}^{h-t-1} (4c)^k$$
$$\leq (4c)^{h-1} + (4c)^t \cdot 2(4c)^{h-t-1} \leq 3 \cdot (4c)^{h-1}.$$

The first inequality is since at any time $t$ the optimal solution opens a single facility and there are $(4c)^t$ clients that are at distance $\sum_{k=0}^{h-t-1}(4c)^k$ from the facility that is located in a leaf of their subtree.

We next observe that if in round $t = 1, \ldots, h-1$ if it holds that $x_L^t + x_R^t \leq 1/2$, then the algorithm pays a cost of at least $c \cdot \text{OPT}^t$. To see this, note that if this condition holds then the set of clients at time $t$ must be matched fractionally to extent at least $1/2$ to a facility outside the subtree of $u^t$. The service cost of this fraction is at least twice the weight of the edge between $u^t$ and its parent, and hence the algorithm's service cost for these $(4c)^t$ clients at time $t$ is at least

$$\frac{1}{2} \cdot (4c)^t \cdot 2 \cdot (4c)^{h-t} = (4c)^h > c \cdot 3(4c)^{h-1} = c \cdot \text{OPT}^t.$$

Finally, we can conclude that any algorithm that maintains an approximation better than $c$ with respect to $\text{OPT}^t$ must open at each time $t = 1, \ldots, h-1$ at least a $1/2$ fraction of a facility in the subtree of $u^t$. By the construction of the adversarial sequence, it must also open at least a $1/4$ facility in the subtree in which there are no additional clients at time $t+1$. In total, the algorithm opens at least $\frac{h-1}{4} = \Omega(h) = \Omega(\min\{\log |F|, \log_c \Delta\}$ facilities, where we observe that $|F| = 2^h$ and $\Delta = O(4c^h)$. When the clients at time $t = h$ leave, the optimal solution does not have any facilities and

therefore the algorithm also must remove all its facilities. Hence, we may repeat the phase/sequence again by initiating a new client at the root.

**A lower bound for the fully-dynamic $k$-median problem.** Assume that there exist a fractional algorithm that maintains at all times at most $k$ centers with an approximation ratio strictly less than $c$ with respect to the optimal solution at time $t$ (we later extend this bound to a scenario in which the algorithm is allowed to open $b \cdot k$ facilities). We prove that such an algorithm has recourse competitiveness $\Omega(\min\{\log n, \log_c \Delta\})$ with respect to an algorithm that may maintain a 2-approximate solution. That is, the algorithm is paying $\Omega(\min\{\log n, \log_c \Delta\}) \cdot \text{OPT}_{\text{REC}}^2$ Our bounds again hold even when $k = 1$.

The metric space and the client sequence are identical to those in the $k$-center instance above. This time, we observe that in each phase, opening a single center on the final $v$ of the sequence is at most a 2-approximation with respect to $\text{OPT}^t$ for all time steps $t$ in the phase simultaneously. (For every $t$, the optimal solution is to open a center on one of $u_L^t$ or $u_R^t$.) The cost of this solution at any round $t = 0, \ldots, h-1$ is

$$2 \cdot \text{OPT}^t = 4 \cdot \sum_{k=0}^{h-1-t} (4c)^k \leq 4 \cdot (4c)^{h-1-t}.$$

At time $t = h$, we have $\text{OPT}^h = 0$.

On the other hand, we also observe that if at time $t = 1, \ldots, h-1$ the algorithm has $x_L^t + x_R^t \leq 1/2$, then the algorithm's solution has cost at least $2c \cdot \text{OPT}^t$. To see this, note that the two clients at $u_L^t$ and $u_R^t$ must be matched fractionally to extent at least $1/2$ to a center outside the subtree rooted at $r^t$. Hence the cost paid by the algorithm is at least four times the cost of the edge between $r^t$ and its parent $r^{t-1}$, which is at least

$$\frac{1}{2} \cdot 4 \cdot (4c)^{h-t} > c \cdot 4(4c)^{h-t-1} \geq 2c \cdot \text{OPT}^t.$$

We conclude as before. Any algorithm maintaining an approximation better than $2c$ with respect $\text{OPT}^t$ must have at each time $t = 1, \ldots, h-1$ at least a $1/2$ fraction of a center open in the subtree of $r^t$. Since the algorithm can open at most one center at any given time, and by construction of the adversarial sequence, at every time $t = 0, \ldots, h-1$, the algorithm has more fractional mass in the child of $r^t$ that is not $r^{t+1}$, the algorithm must open at least $1/4$ of a center to maintain a $2c$-approximation. Hence the algorithm must open a total of $\Omega(h)$ centers over the course of the phase, where we observe that size of the metric is $n = 2^h$ and the aspect ratio is $\Delta = O((4c)^h)$. This sequence may be repeated.

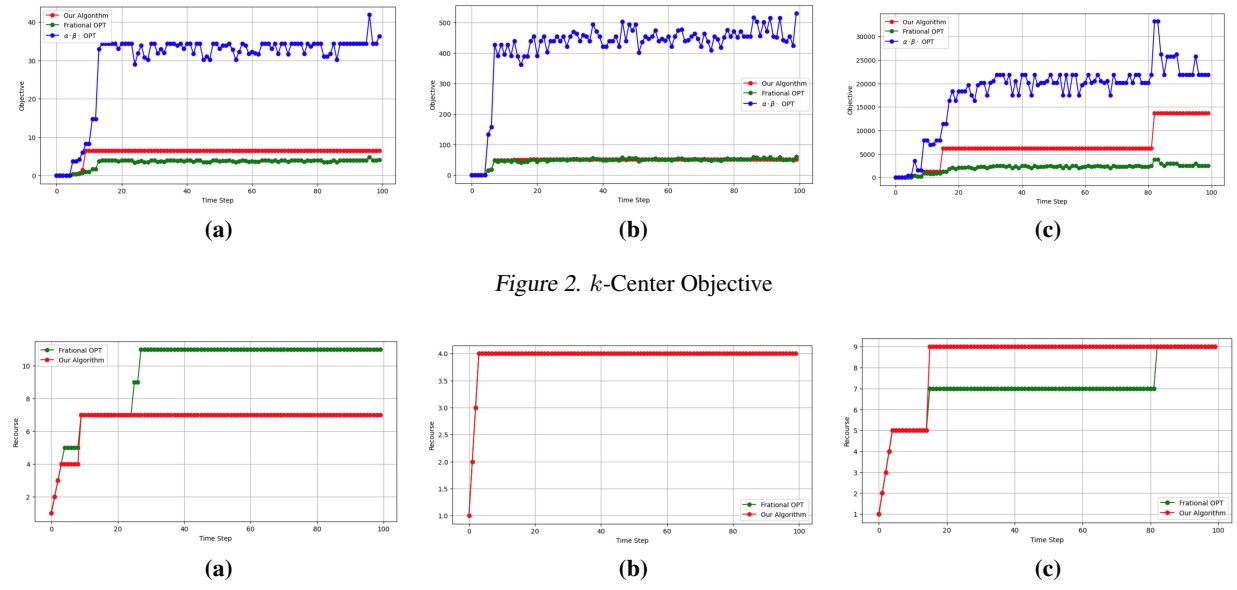

Figure 2. $k$-Center Objective

Figure 3. $k$-Center Recourse

Finally, suppose again that the algorithm is allowed to maintain a total mass of $b \cdot k = b$ facilities at each time step. Then, at the beginning of each phase, there exists a subtree of height $h' = \Omega(h - \log_2 b)$ with total initial mass of at most $1/4$. The adversary can restrict its sequence to this subtree and the algorithm, again, must open $1/4$ of a facility in each level of the tree, paying a total opening cost of $\Omega(h - \log_2 b)$ during the phase. Overall, the lower bound on the recourse competitiveness is $\Omega(\min\{\log n, \log_c \Delta\} - \log b)$.

## D. Experiments continued

Each column below corresponds to one of these datasets ((a) Glass, (b) Wine, (c) Airfoil), and the data was streamed online as follows for $T = 100$ times steps. At each time $t$, either a new data point was inserted with probability $9/10$, or an existing data point was deleted with probability $1/10$. We set $k = 4$ (for $k$-center and $k$-median), $\beta = 3/2$ and $\epsilon = 1/4$.

In the objective plots, we plot the objective of our online algorithm over time (red), alongside the value of the fractional optimum $\text{OPT}^t$ at each point in time (green), as well as $\alpha \cdot \beta \cdot \text{OPT}^t$ (blue), which our theorems guarantee is an upper bound on the algorithms objective. In the recourse plots, we plot the recourse of our online algorithm over time (red), alongside the optimum fractional recourse without resource augmentation (green), i.e. exactly $k$ centers, exactly $\beta$ approximation. Finally, in the last set of plots, we track the number of centers opened by our $k$-center and $k$-median algorithms over time.

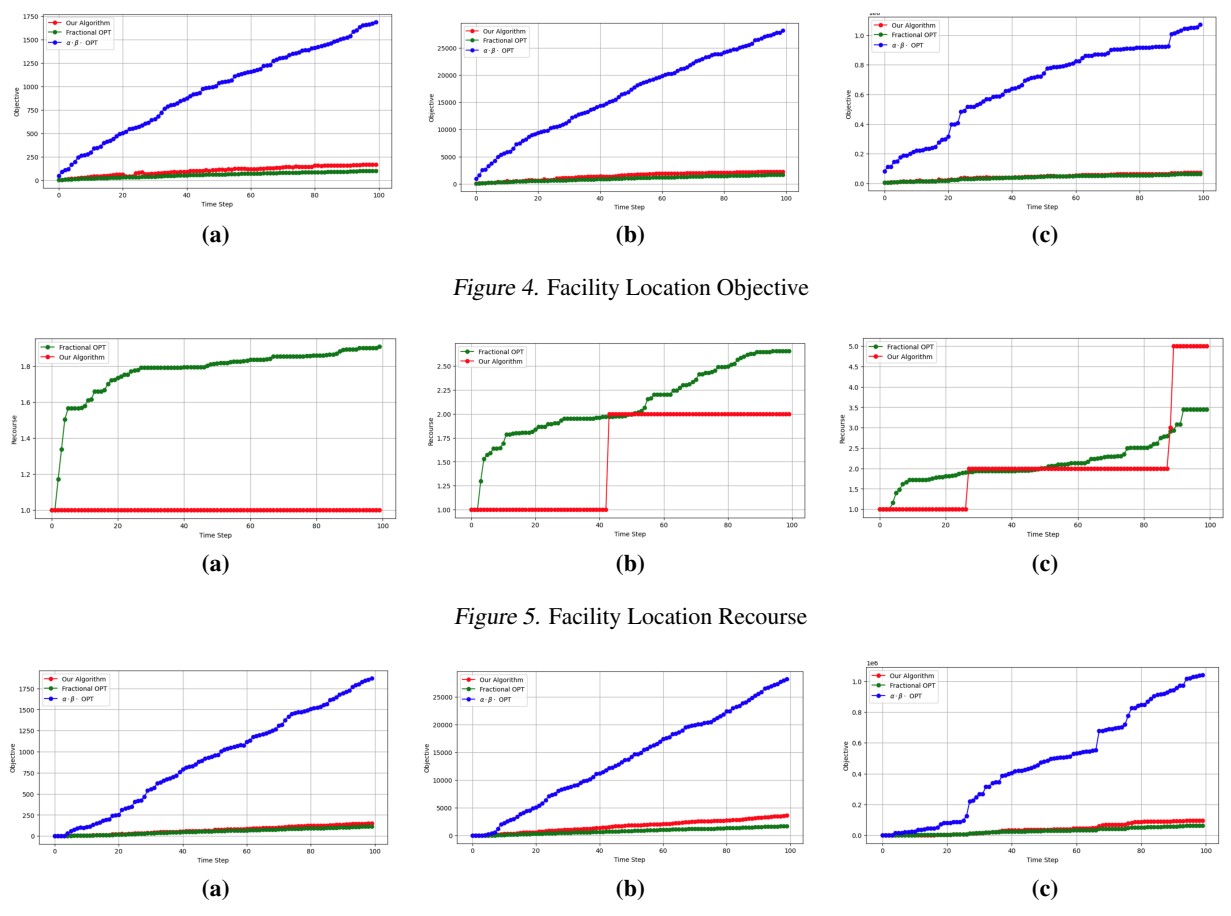

*Figure 4.* Facility Location Objective

*Figure 5.* Facility Location Recourse

*Figure 6.* $k$-Median Objective

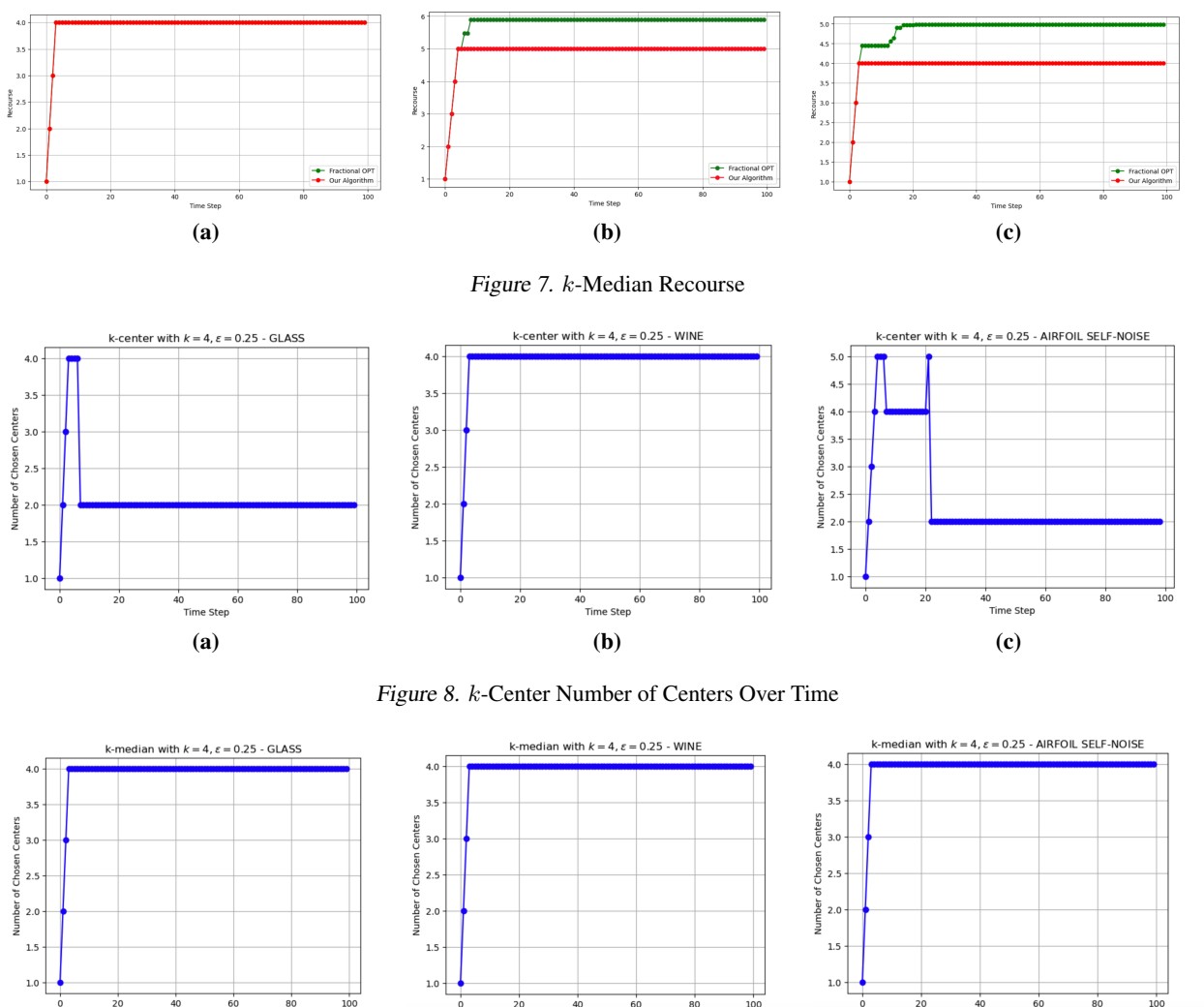

*Figure 7. k-Median Recourse*

*Figure 8. k-Center Number of Centers Over Time*

*Figure 9. k-Median Number of Centers Over Time*

