# OpenReview forum: "Competitively Consistent Clustering"
_ICML.cc/2025/Conference — ICML 2025 poster_

### Official Review · Reviewer_Bgbi · 2025-03-10

**Overall Recommendation:** 4

**Summary:**

When clustering dynamic data (with insertions and/or deletions), consistency of updated solutions is a  concern that can be more relevant that optimal cost in practice. This work studies fully dynamic clustering algorithms for k-center, facility location, and k-median with competitive recourse. For fully dynamic inputs, the pathological example that required linear recourse inserts and deletes the same $O(1)$ points over and over again so that the value of an optimal solution oscillates between two magnitudes. Let $OPT^\\beta_R$ be the recourse of an optimal offline algorithm with multiplicative cost approximation guarantee $\\beta$ for the problem at hand, and $\\Delta$ be the aspect ration of the input. The authors obtain the following dynamic algorithms:

1. For k-center, a bi-criteria approximation that uses $(1+\\alpha)k$ centers and has multiplicative $O(\\beta)$ cost approximation with recourse $O(\\log n \\log \\Delta / \\epsilon^2) OPT^\\beta_R$.
2. For facility location, it maintains a multiplicative $O(\\beta)$ cost approximation with recourse $O(\\log |F| \\log \\Delta) OPT^\\beta_R$, where $F$ is the possible set of centers.
3. For k-median, a bi-criteria approximation that uses $O(k)$ centers and has multiplicative $O(\\beta)$ cost approximation with recourse $O(\\log n \\log \\Delta) OPT^\\beta_R$.

The authors also show that at least one of the logarithmic dependencies is required and conduct experiments in the appendix.

## update after rebuttal

The authors addressed my questions and could clarify my doubts. I adjusted my scored accordingly.

**Claims And Evidence:**

All claims are backed by proofs.

**Essential References Not Discussed:**

None identified.

**Experimental Designs Or Analyses:**

The extent of the experimental setups (3 datasets, one choice for k) is a bit limited, but the setup itself is appropriate.

**Methods And Evaluation Criteria:**

Rigorous theoretical analysis combined with experiments is a very reasonable approach.

**Other Comments Or Suggestions:**

It seems appropriate to mention in the abstract that the algorithms for k-center/k-median actually compute bi-criteria approximations.

Whenever $k'$, it may be worth to consider if it's useful to mention its best known value as derived from the analysis in appendix A.

**Other Strengths And Weaknesses:**

The paper introduces algorithmically nice and simple, but analytically non-trivial rounding schemes on top of a known algorithm for maintaining dynamic fractional solutions by Bhattacharya et al. By using competitive recourse, the authors show that one can circumvent the trivial worst-case bounds for fully dynamic algorithms and optimal close(r) to optimal recourse.

On the other hand, bi-criteria solutions that use $O(k)$ centers that are compared to offline algorithms that use $k$ centers open a new gap. E.g., consider k-center and $k+1$ input points (or clusters) at large distance. While an optimal solution needs to pay the distance at least one, an algorithm that uses $k+1$ centers can achieve cost 0. Nevertheless, not using exactly $k$ but $O(k)$ centers is a concern that often lies mostly on the theory side of the problem.

**Questions For Authors:**

1. Can you add plots that show the actual number of centers that the dynamic algorithm opened?

2. Can the k-median results be extended to k-clustering (e.g., k-means)?

**Relation To Broader Scientific Literature:**

Consistent clustering of dynamic data is an emerging topic in theory, and inherently important in many practical applications.

**Theoretical Claims:**

Checked plausibility of proofs in the main part and skimmed through the appendix for completeness.

---

> ### Author Rebuttal · Authors · 2025-03-29
>
> Thanks for the thorough and very positive review.
>
> We will mention in the abstract that the algorithm are bi-criteria. Indeed, theoretically, the gap between a solution that uses k and k+1 centers may be large. However, we believe this rarely happens in practical instances.
>
> The rebuttal platform does not allow for us to upload new plots, but we have added the plot of # centers opened as a function of time to our manuscript. In both our k-center and k-median experiments, the algorithm almost always opens no more than k centers.
>
> We did not think about the k-means objective. This is indeed a great and interesting future research problem.

---

### Official Review · Reviewer_G5B5 · 2025-03-11

**Overall Recommendation:** 3

**Summary:**

This paper studies fully dynamic consistent clustering, specifically focusing on the $k$-center, facility location, and $k$-median problems. Previous work has focused on algorithms maintaining solutions close to optimal while minimizing recourse—the number of changes to centers over time. The key innovation here is the consideration of a "beyond worst-case" scenario. The authors design algorithms that maintain approximate solutions with recourse bounded competitively against the minimal recourse achievable by an offline algorithm.

The main technique builds on the Positive Body Chasing framework recently introduced by Bhattacharya et al. [FOCS 2023], which provides fractional solutions to clustering problems. The paper's primary contribution is developing rounding methods to convert fractional solutions to integral ones while preserving approximation and recourse guarantees.

**Claims And Evidence:**

Proofs support the theorem statements. The paper explains high-level proof techniques quite clearly, and to my best comprehension, the proofs presented in the main body and appendix are correct.

However, there is one important concern about consistency in the problem definition. The paper considers clustering formulations ($k$-center, facility location, and $k$-median), typically defined with at most $k$ centers allowed. Yet, in Theorem 1.1, the proposed algorithms use $(1+\varepsilon) \cdot k$ or even $O(k)$ centers, where $\varepsilon \in (0, 1]$. This contradicts standard definitions in the literature, where dynamic algorithms strictly maintain exactly $k$ centers (e.g., Łącki et al., 2024; Bhattacharya et al., 2024). Allowing extra centers makes the problem strictly easier, possibly trivializing it, as one could achieve approximation ratios smaller than 1 for the $k$-center problem by opening additional centers. I realise this issue is acknowledged briefly in the conclusion, but it should be justified and explained more clearly.

**Essential References Not Discussed:**

N/A

**Experimental Designs Or Analyses:**

From the text the experimental design seems sound and valid. I wanted to check the code as well, but unfortunately no supplementary material was provided.

**Methods And Evaluation Criteria:**

The experiments compare the proposed method against the optimal fractional solutions and the theoretical bounds stated in the theorems, which is appropriate. However, the paper lacks experimental comparisons to previous methods from the literature, such as "Efficient and Stable Fully Dynamic Facility Location" by Bhattacharya et al. [NeurIPS 2022], which implemented a related algorithm for dynamic facility location.

**Other Comments Or Suggestions:**

1. line 036, right column: Can use \citet to reference Lacki et al. (2024). Also on line 085, right column, among other places.
2. In Theorem 1.1, $\varepsilon$ is not defined.

**Other Strengths And Weaknesses:**

S1) I think the overall proof techniques are quite nice. Combining the Positive Body Chasing framework with the different rounding mechanisms is nice, and I could see it having potential future impact.

**Questions For Authors:**

Q1. In the introduction you describe the problem as: $k$-center, in which we can open _at most_ $k$ centers (...). However, in the statement of Theorem 1.1, your maintained solution uses $(1+\varepsilon) \cdot k$ centers. Since $\varepsilon \in (0, 1]$, this means your solution uses potentially $2k$ clusters. However this is different than the formal problem description (see the Claims and Evidence section above for more elaboration). Could you please clarify this, in case I misunderstood something.

Q2. Could you please elaborate on the update time of each algorithm? Your proposed method has low recourse, but I'm curious about the guarantees on the running time of each update.

**Relation To Broader Scientific Literature:**

This paper contributes to the growing body of work on dynamic clustering and online approximation algorithms. Previous work on fully dynamic clustering focused on absolute recourse, e.g., Łącki et al. (2024) and Bhattacharya et al. (2024). In contrast, this work introduces the notion of competitively consistent clustering, aligning itself with recent trends in the broader literature that advocate competitive analysis against optimal offline recourse. This competitive recourse perspective originates from Bhattacharya et al. [FOCS 2023], and the current paper is (to the best of my knowledge) the first to effectively extend this framework to classical clustering objectives through tailored rounding schemes.

**Theoretical Claims:**

I checked all proofs and found no issues.

---

> ### Author Rebuttal · Authors · 2025-03-29
>
> Thanks for the thorough review. Indeed, in the statement of Theorem 1.1, the result for the k-center is for every \eps\in(0,1/2). We will add this to the statement. Our algorithms are with 'resource augmentation' which is very common in competitive analysis (e.g. in caching, network routing, scheduling, see Chapter 4 of Beyond the Worst-Case Analysis of Algorithms by Tim Roughgarden for a survery), and Theorem 1.1 explicitly states this as an assumption. As reviewer 4 suggested, we will emphasize this more (also in the abstract).
> One may also think equivalently about resource augmentation as allowing our algorithm to use at most k centers, but comparing our solution to an all powerful offline solution (that knows the future), but is only allowed to use slightly less than k centers.
>
> As Reviewer 4 (Bgbi) notes, giving an algorithm that opens precisely k instead of O(k) centers is a worthy open question, but a mostly theoretical one: in practice the parameter k is usually selected heuristically, and users of such algorithms are often equally happy with a clustering with k' = O(k) clusters so long as it still explains the data well, doesn't overfit, etc. Finally, we have added experiments to our paper that show that in practice on our test instances, our k-center and k-median algorithms tends to open at most k centers anyway.
>
> About empirical comparison to prior work: in our comments to Reviewer 1 (iX2S), we argue that the fractional OPT against which we compare our algorithm is a the strongest possible dynamic benchmark (up to the constant slack factor of beta).
>
> About the update time: Our main concern in this paper is the recourse. Note that already the update time for maintaining the fractional solution (Bhattacharya et al. [FOCS 2023]) may be polynomial. A naive implementation of our rounding algorithm requires poly(n,k) update time, where n is the number of clients at the time step. We believe such an update can be done quite efficiently, but still at least in linear time. Designing an algorithm with competitive recourse and also fast update time is indeed an interesting distinct research question.

---

> > ### Comment · Reviewer_G5B5 · 2025-04-02
> >
> > Thank you for your response and for clarifying the significance of the theoretical results - please do emphasise the fact that your algorithms are bicriteria in the abstract and introduction. I still think the experiment section lacks a comparison to existing literature.
> >
> > I've updated my score accordingly.

---

### Official Review · Reviewer_m98A · 2025-03-12

**Overall Recommendation:** 4

**Summary:**

This paper considers dynamic clustering problems, including dynamic k-center, facility location, and k-median.  The goal is to maintain constant factor approximation with small recourse (the total changes made to the solution).

All existing works in this problem obtain an absolute recourse guarantee, namely, they prove a bound on the total number of recourse. Authors observe that such worst-case bound often occurs in a pathological situation. However, in most natural cases, the optimal offline recourse can often be made much smaller. This motivates the study of algorithms with competitive recourse guarantees, which are popular goals in the dynamic setting.

The authors design algorithms with competitive recourse for all three clustering problems studied. The competitive ratio is logarithmic in the metric spread and the number of candidate centers. Specifically, their algorithm maintains a bicriteria approximation solutions for $k$-center and $k$-median, and a constant factor approximation for facility location, both with logarithmic competitive ratio.

Technically, the algorithm builds on a recent chasing positive body framework, with the development of certain rounding techniques to satisfy the recourse guarantees.

Moreover, a lower bound is proved to show that the logarithmic competitive ratio is the best one can hope for, and experiments have been conducted to support the claims.

**Claims And Evidence:**

The claims have been proved rigorously.

**Essential References Not Discussed:**

No

**Experimental Designs Or Analyses:**

Authors have implemented their algorithms to verify their claims.

**Methods And Evaluation Criteria:**

Yes

**Other Comments Or Suggestions:**

I have not found significant typos. The paper is written well.

**Other Strengths And Weaknesses:**

strength: the result is novel with theoretical depth.

weakness: Although the experiment has been conducted, it is mostly for the aim of verifying the recourse guarantee. It might be good to test against some natural dynamic algorithms as benchmarks to show practical value. So far, I think the main value of this result is on the theory side.

**Questions For Authors:**

What is the updated time of your algorithm?

**Relation To Broader Scientific Literature:**

I think the problem studied is important in algorithmic machine learning. The main results of this paper may be interesting to anyone interested in clustering.

**Theoretical Claims:**

I have not checked all the proofs in the appendix.

---

> ### Author Rebuttal · Authors · 2025-03-29
>
> Thanks for the thorough and very positive review. In our comments to Reviewer 1 (iX2S), we argue that the fractional OPT against which we compare our algorithm is a the strongest possible dynamic benchmark (up to the constant slack factor of beta). See our comments to Reviewer 3 (G5B5) about the update time.

---

### Official Review · Reviewer_iX2S · 2025-03-16

**Overall Recommendation:** 4

**Summary:**

This paper studies fully dynamic clustering algorithms with competitive recourse guarantees. The authors focus on three classic clustering problems: k-center, facility location, and k-median. In the fully dynamic setting, given a metric space with n data points, a different set of points is chosen as clients at each time step. Then, the algorithm is required to maintain a sequence of clustering solutions over steps such that the clustering cost is approximately optimal in every step while minimizing the recourse, which is the total changes in the center set across steps.

Let $\Delta$ be the aspect ratio of the metric space and $F$ be a subset of facilities in the space. For every $\beta \geq 1$, let $OPT_\beta$ be the optimal recourse of an offline solution that maintains $\beta$ approximation.

They provide algorithms for these problems which satisfy: for every $\beta \geq 1$, and $\epsilon \in (0,1/2)$,

- For k-center, it maintains an $O(\beta)$ approximate solutions with $(1+\epsilon)k$ centers while ensuring recourse at most $O(1/\epsilon^2 \cdot \log |F| \log \Delta)OPT_\beta$.

- For facility location, it maintains an $O(\beta)$ approximate solutions while ensuring recourse at most $O_{\epsilon}(\log |F| \log \Delta)OPT_\beta$.

- For facility location, it maintains an $O(\beta)$ approximate solutions with $O(k)$ centers while ensuring recourse at most $O_{\epsilon}(\log |F| \log \Delta)OPT_\beta$.

They also provide the lower bound which shows that for any $\beta \geq 1$, any randomized algorithm that maintains an $O(\alpha \cdot \beta)$ solution requires recourse at least $\Omega(\min(\log |F|, \log_\alpha \Delta)) OPT_\beta$, moreover, for k-center and k-medians with even O(k) centers.

They achieve the upper bound by leveraging the Positive Body Chasing framework, which provides fractional solutions that they round efficiently while preserving approximation and recourse guarantees. The paper presents both theoretical results with upper and lower bounds on recourse and experimental validation on UCI datasets, showing that the proposed methods significantly outperform the worst-case guarantees in practice.

**Claims And Evidence:**

Yes, the paper's claims are well-supported by both theoretical analysis and empirical results.
They provide both upper bound and lower bound for the problems and run experiments to validate their algorithm on real-world datasets.

**Essential References Not Discussed:**

No. The paper covers the necessary references.

**Experimental Designs Or Analyses:**

Yes, I checked the validity of the experiments in the appendix.

**Methods And Evaluation Criteria:**

The methods and evaluation criteria are appropriate.

They utilize the Positive Body Chasing to get a fractional online solution, and then provide rounding techniques carefully designed to balance approximation quality and recourse minimization. The experiments use benchmark datasets from the UCI repository. The evaluation criteria are reasonable and well-justified, with an emphasis on both approximation quality and recourse efficiency.

**Other Comments Or Suggestions:**

Minors:

- Line 282, Proof of Lemma 3.3, I think $OPT^t$ should be $r_j$.
- Line 355, Proof of Theorem 3.4, there seems to be a missing sum over all time steps t = 0,1,..., T.
- In Theorem 1.1, What is the range of the parameter $\epsilon$? Also, it seems from the later analysis, for facility location and k-medians, the approximation factor for recourse should have a dependence on $\epsilon$ as well?
- In Remark 3.2 and 3.5, I can get the intuition of the analysis. But, if there is space, I think it would be better to include a detailed analysis for one of them in the main body?
-Maybe some experiments can be included in the main body, one or two figures due to the remaining space in the paper?

**Other Strengths And Weaknesses:**

Strength:
- The paper proposes these interesting fully-dynamic clustering problems.
- The paper provides interesting algorithms and gives theoretical analysis of the approximation on clustering cost and the total recourse.
- The paper also provides the lower bound on the recourse approximation factor for the fully dynamic clustering algorithm by constructing an interesting instance with HST metric.

**Questions For Authors:**

- In the experiments, you mainly test your algorithms, the fraction optimal solution, and the upper bound given by the analysis. Would it be interesting to compare your algorithm with some other traditional clustering algorithms which are static and do not care about the recourse at all (just recompute the solution at every time step)?

**Relation To Broader Scientific Literature:**

The paper is well-connected to broad research topics including consistent algorithms, clustering algorithms, and online and dynamic algorithms.

**Theoretical Claims:**

Yes, I checked the proof of all upper bounds and lower bounds in the paper and appendix. While I only briefly checked the positive body chasing framework to achieve the fractional solution in appendix A.

---

> ### Author Rebuttal · Authors · 2025-03-29
>
> Thanks for the thorough and very positive review. We fixed the comments, and will try to include a more formal proof of the remark. In Theorem 1.1 in the facility location and k-median, we already lose a constant in the cost (or the number of servers). Hence, one may simply use \eps=1. We will add a comment on this.
>
> A static algorithm that does not care about recourse, and recomputes the solution from scratch at every time may have a very large recourse. Consider, for example, a dynamic 1-center problem in which there exists a single point that is a good center for all time steps. An offline algorithm that recomputes the solution may fail to detect this point and keep using a different point at every time step. We claim that the benchmark we use in our experiments is more stringent/informative. We compare against the fractional OPT (up to a constant slack factor beta which we set to 1.5); up to this slack factor, this is better than any offline benchmark. Furthermore, of all possible fractional OPT solutions, this is the one with the best recourse possible.

---

### Decision · Program_Chairs · 2025-05-01

**Decision:**

Accept (poster)

**Comment:**

The paper introduces competitively consistent clustering for fully-dynamic k-center, facility location, and k-median problems, aiming to maintain approximately optimal solutions while minimizing recourse compared to the optimal offline recourse. The authors design algorithms using a reduction to the positive body chasing framework, providing fractional solutions which are then rounded. Nearly matching logarithmic lower bounds are also provided.

This work proposes interesting solutions for fully-dynamic clustering problems and so it would be a good addition to the conference program. The techniques are quite nice, particularly the reduction to positive body chasing and the subsequent online rounding methods. Furthermore, the paper's claims are well-supported by both theoretical analysis and empirical results. A minor weakness is that although the experiment has been conducted, it is mostly for the aim of verifying the recourse guarantee, and perhaps could be expanded slightly to show performance regarding the clustering objective itself more directly.